# SleepLM: Natural-Language Intelligence for Human Sleep

**Zongzhe Xu** [1]  **Zitao Shuai** [1]  **Eideen Mozaffari** [1]  **Ravi S. Aysola** [1]  **Rajesh Kumar** [1]  **Yuzhe Yang** [1]

## Abstract

We present `SleepLM`, a family of sleep-language foundation models that enable human sleep alignment, interpretation, and interaction with natural language. Despite the critical role of sleep, learning-based sleep analysis systems operate in closed label spaces (e.g., predefined stages or events) and fail to describe, query, or generalize to novel sleep phenomena. `SleepLM` bridges natural language and multimodal polysomnography, enabling language-grounded representations of sleep physiology. To support this alignment, we introduce a multilevel sleep caption generation pipeline that enables the curation of the *first* large-scale sleep-text dataset, comprising over 100K hours of data from more than 10,000 individuals. Furthermore, we present a unified pretraining objective that combines contrastive alignment, caption generation, and signal reconstruction to better capture physiological fidelity and cross-modal interactions. Extensive experiments on real-world sleep understanding tasks verify that `SleepLM` outperforms state-of-the-art in zero-shot and few-shot learning, cross-modal retrieval, and sleep captioning. Importantly, `SleepLM` also exhibits intriguing capabilities including language-guided event localization, targeted insight generation, and zero-shot generalization to unseen tasks. To support reproducibility and future work, we open-source the captioning pipeline, pretrained checkpoints, and the model architectures at `https://github.com/yang-ai-lab/SleepLM`.

## 1. Introduction

Humans move between two distinct states: the *waking* life, structured by perception and language; and *sleep*, expressed through dense and continuous physiology. Sleep is not simply rest; it is a dynamic orchestration shaped by interactions among brain oscillations, cardiac regulation, and respiratory rhythms (Berry et al., 2012; Brink-Kjaer et al., 2022; Eban-Rothschild et al., 2018). Polysomnography (PSG) captures this narrative through synchronized channels (e.g., EEG, ECG, EMG, EOG, respiration), offering a high-resolution view into human health and encoding rich biomarkers for cardiovascular, neurological, and metabolic function (Addo et al., 2024; Vallat et al., 2023; Yang et al., 2022).

As with waking experience, making sense of sleep requires a mapping from ***physiology*** to ***language***. The same PSG pattern can be described from low-level signal shifts (e.g., *"EEG delta power increases and heart rate slows"*) to high-level sleep states (e.g., *"transition into deep sleep"*). Learning this translation and alignment is the key to interpreting and interacting with high-frequency sleep physiology through actionable language descriptions, which is essential for democratizing sleep health (Cosentino et al., 2024), enabling personalized monitoring (Mogavero et al., 2025) and advancing clinical analysis beyond fixed categorization.

Yet, current computational methods do not meet this need. On one hand, existing machine learning models for sleep are predominantly *discriminative* and confined to *closed* label spaces (e.g., sleep stages or events) without the capacity for open-ended description or reasoning (Bahrami & Forouzanfar, 2022; Nie et al., 2025; Perslev et al., 2021). On the other hand, while Large Language Models (LLMs) excel at generative tasks, they are inherently ill-equipped to handle the high-dimensional, continuous nature of physiological data (Li et al., 2026): Fig. 1 shows the poor zero-shot performance of Gemini 2.5 Pro (Comanici et al., 2025) and DeepSeek-R1 (Guo et al., 2025) (details in Sec. 5), leaving the rich narrative of sleep largely inaccessible to modern generative AI.

To fill the gap, we present `SleepLM`, to our knowledge, the first family of sleep-language foundation models that unlocks meaningful interpretation of raw sleep signals and enable novel sleep applications through natural language. The efficacy of `SleepLM` represents a paradigm shift from *categorizing* sleep to *describing* and *interacting* with it, enabled by three key innovations: ❶ A hierarchical, automated captioning pipeline that generates structural descriptions for each sleep epoch, spanning global semantic summaries, local fine-grained details, and channel-specific characteristics;

---

[1]University of California, Los Angeles. Correspondence to: Yuzhe Yang <yuzhey@ucla.edu>.

*Proceedings of the 43rd International Conference on Machine Learning*, Seoul, South Korea. PMLR 306, 2026. Copyright 2026 by the author(s).

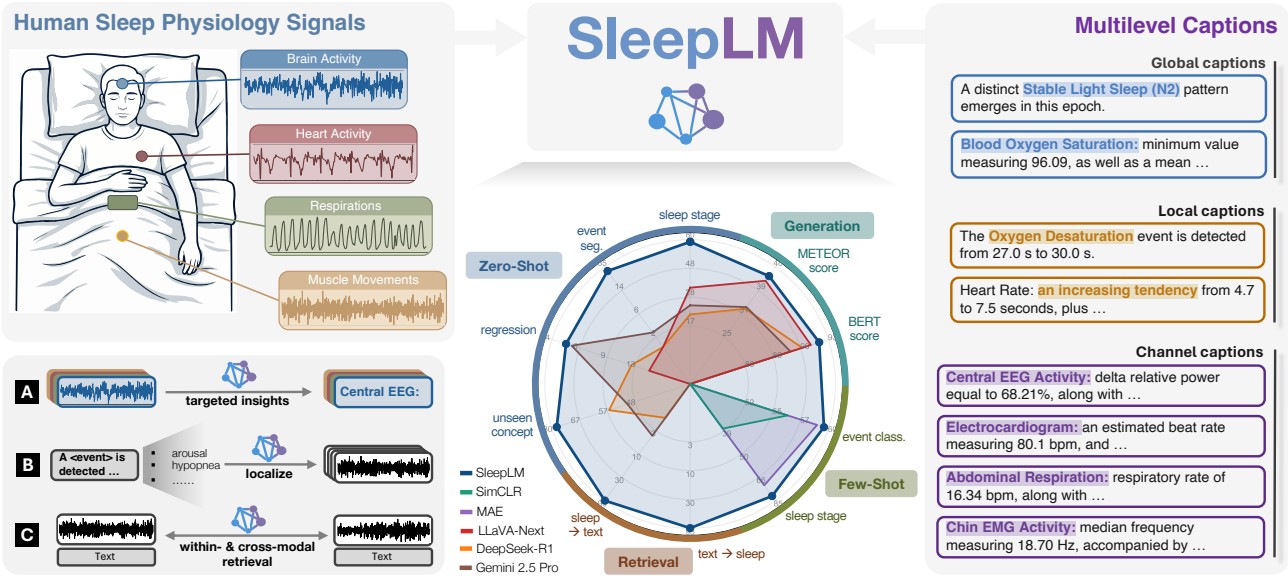

Figure 1. **Sleep-language foundation models** (`SleepLM`). We present a comprehensive study using over 100K hours of multimodal sleep PSG data from over 10,000 individuals. We design a multi-level captioning pipeline that captures PSG information at different temporal and semantic granularities. Across a wide range of downstream tasks and evaluation settings, `SleepLM` consistently outperforms state-of-the-art LLMs and VLMs. In addition to its predictive capabilities, `SleepLM` also enables: **(A)** targeted and controlled insights generation, **(B)** language-guided event localization, and **(C)** within- and cross-modal zero-shot retrieval (details in Sec. 5).

❷ The curation of the *largest* paired sleep-text dataset to date, comprising over 100,000 hours of data from more than 10,000 individuals, enabling the learning of robust, generalized representations of human sleep; and ❸ `ReCoCa`, a novel and generic multimodal pretraining architecture that utilizes a compound objective (contrastive alignment, caption generation, and signal reconstruction) to support scalable learning of joint language and physiological time-series data.

To rigorously evaluate `SleepLM`, we benchmark it against state-of-the-art (SOTA) methods across diverse, real-world sleep understanding tasks. Extensive experiments verify that `SleepLM` not only achieves superior performance on established tasks, but also enables new capabilities such as controllable insight generation and zero-shot generalization to novel clinical concepts. Our contributions are as follows:

- We introduce a multilevel captioning pipeline for sleep data, yielding the largest sleep-text dataset to date with over 100,000 hours of data from over 10,000 people.
- We design `SleepLM`, the first family of sleep-language foundation models that enables diverse sleep capabilities and interaction through natural language.
- We present `ReCoCa`, a unified and generic multimodal pretraining architecture for joint learning over language and physiological time-series data.
- Extensive experiments across real-world sleep tasks verify the superiority of `SleepLM` against SOTA methods, and reveal emergent capabilities including controllable insight generation and generalization to unseen concepts.

## 2. Related Work

**Sleep Foundation Models.** Sleep research has begun adopting large-scale pretraining for physiological signals, producing strong representations for discriminative tasks such as sleep staging and event detection (Carter & Tarassenko, 2024; Shuai et al., 2026; Zhang et al., 2022). Current methods focus on self-supervised learning for biological signals, including inter-channel contrastive learning (Thapa et al., 2024), masked reconstruction (Pandey et al., 2024), and predictive coding for long-range sleep structure (Eldele et al., 2023). However, existing sleep foundation models remain unimodal and optimized for classification, limiting their ability to generate descriptions or handle new concepts. In contrast, our work aligns raw PSG with *natural language* to support open-ended description, controllable reporting, and zero-shot generalization beyond fixed label sets.

**Time Series Foundation Models.** Time series modeling has advanced with foundation models that generalize across many downstream settings, including weather forecasting and financial data (Ansari et al., 2025; Cohen et al., 2025; Woo et al., 2024). Recent work also studies how to represent temporal structure effectively, including tokenization and parameterization (Nie et al., 2023), pretraining objectives (Zeng et al., 2023), architectural choices (Das et al., 2024), and embedding strategies (Ansari et al., 2024). Although physiological signals share the same temporal form, they differ in sampling rate, recording length, and fixed channel structure. In contrast, we adapt these time-series design principles to PSG by explicitly accounting for multi-channel

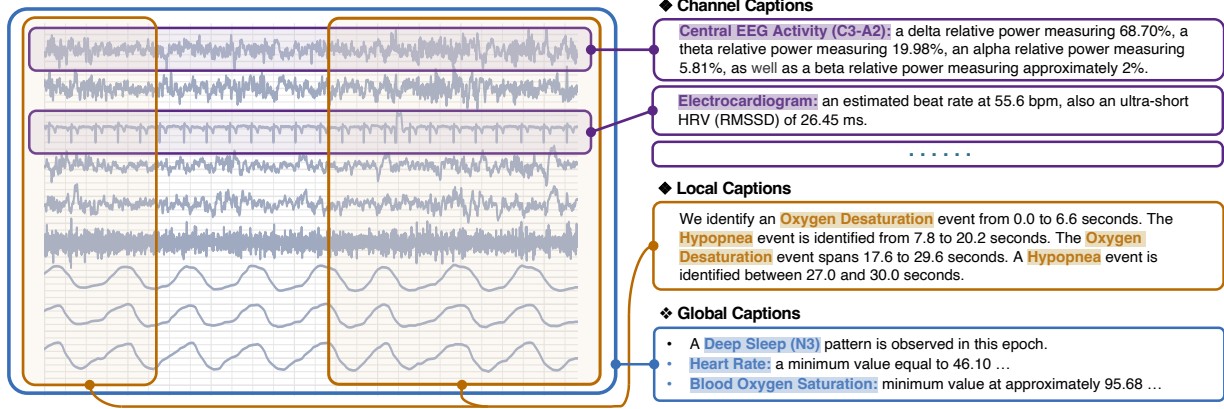

*Figure 2.* **Multilevel sleep captioning pipeline.** We generate three complementary levels of captions from each PSG window: (1) Channel captions summarize modality-specific clinically relevant statistical features commonly used in manual scoring; (2) Local captions capture temporal semantics such as transient morphological changes and sleep event onsets and durations; (3) Global captions describe high-level physiological states such as sleep stage and overall cardiac and respiratory conditions. Example captions are provided in Appendix E.1.

physiology and long, dense recordings, and by pairing signals with language for interpretability and interaction.

**Multimodal Language Models.** Connecting non-text modalities with language models is now central in multimodal learning. In vision, CLIP (Radford et al., 2021) showed that contrastive alignment between images and text enables strong zero-shot transfer, and CoCa (Yu et al., 2022) combined contrastive learning with captioning to support both retrieval and generation. These alignment ideas have since been extended to temporal data: OpenTSLM (Langer et al., 2025) adapts language-modality alignment to general time series, while SensorLM (Zhang et al., 2025) pairs wearable signals with automatically generated text to enable natural language interaction with sensor data. We extend this language-signal alignment framework to multi-channel PSG by introducing a sleep-specific captioning pipeline, a large paired sleep-text dataset, and a multimodal pretraining architecture designed for dense physiological time series.

## 3. Human Sleep Captioning at Scale

**PSG Data and Processing.** We source our primary pretraining corpus from the National Sleep Research Resource (Zhang et al., 2018). Specifically, we aggregate five large-scale datasets: SHHS (Quan et al., 1997), MrOS (Blackwell et al., 2011), CFS (Redline et al., 1995), (Rosen et al., 2003), and WSC (Young et al., 2009). For preprocessing, continuous recordings are segmented into non-overlapping 30-second epochs, following standard clinical sleep scoring conventions (American Academy of Sleep Medicine, 2025). We use a standardized 12-channel montage grouped by physiological modality: ❶ *Brain (EEG/EOG):* `C3-A2`, `C4-A1`, `E1-A2`, `E2-A1`. ❷ *Respiration:* `Thorax`, `Abdominal effort`, `Airflow`. ❸ *Cardiac:* `ECG`, `Heart Rate`, `SpO2`. ❹ *Somatic:* `Chin EMG`, `Body Position`. Detailed preprocessing steps are provided in Appendix B.

**Multilevel Sleep Caption Generation.** Foundation models benefit from dense, structured supervision, yet most sleep datasets provide only a single coarse label per 30-second epoch (e.g., stage label such as "N2"), creating a strong information bottleneck (Bilal et al., 2025). To address this, we introduce a ***multilevel*** captioning pipeline that produces hierarchical, text supervision for each PSG epoch. The captions are organized at three complementary granularities:

*Channel **Captions:*** These captions describe signal morphology that is directly observable in the input channels. For each modality (e.g., EEG), we extract clinically used features from standard scoring practice, such as EEG band power and respiratory periodicity and variability. We then render these features into diverse linguistic templates, promoting robustness to language variation while keeping the supervision tightly grounded to the measured signals.

*Local **Captions:*** This level captures within-epoch temporal structure and event localization. We apply trend and peak detection to identify transient changes such as heart-rate accelerations and oxygen denaturation. Beyond event presence, we also indicate onset and offset timestamps within each epoch (Fig. 2), providing fine-grained temporal supervision to enable both event *identification* and *localization*.

*Global **Captions:*** At the highest level, we integrate semantic summaries of the epoch's holistic state, including sleep stage (e.g., "N2", "REM") and global autonomic descriptors. Notably, we derive some of these descriptors from withheld signals (e.g., average heart rate and baseline $SpO_2$) that are excluded from the model input, encouraging inference of these consequences from the remaining biosignals and acting as a masked-prediction proxy task during pretraining.

**Large-Scale Pretraining Sleep-Text Dataset.** Building on the aggregated PSG corpus and the multilevel captioning pipeline, we construct the *first* large-scale paired sleep-text

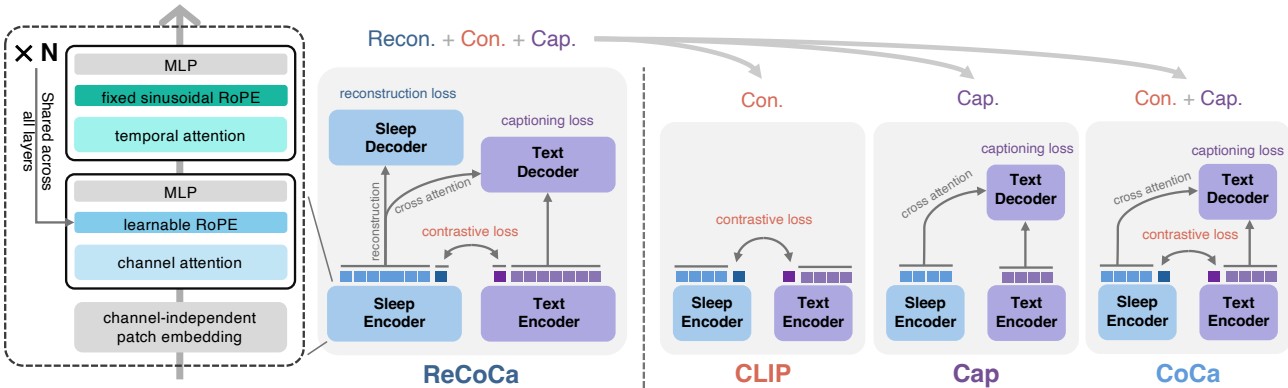

*Figure 3.* **The** `SleepLM` **architecture, pretraining objectives, and variants.** We introduce `ReCoCa`, a generic sleep-language pretraining framework that jointly optimizes signal reconstruction, contrastive alignment, and caption generation for multi-channel PSG. By enabling or disabling components, `ReCoCa` yields common formulations (e.g., CLIP, Cap, CoCa) as special cases.

dataset. In total, the dataset comprises over 100,000 hours of PSG data spanning over 12,000 recording nights from more than 10,000 individuals. Detailed cohort statistics and dataset splits are provided in Appendix B.

## 4. SleepLM

`SleepLM` introduces a generic sleep-language pretraining framework for learning joint representations of sleep PSG and text. `SleepLM` is trained with a compound objective that combines contrastive alignment, caption generation, and signal reconstruction. By enabling or disabling components, this framework instantiates common formulations (e.g., CLIP, Cap, CoCa) as special cases.

**Reconstructive Contrastive Captioner (`ReCoCa`).** We present `ReCoCa`, a unified multimodal pretraining architecture designed for dense physiological time series and forms the default instantiation of the `SleepLM` family (Fig. 3):

*Channel-Specific Sleep Encoder.* PSG channels are not interchangeable: they follow a fixed montage with modality-specific meaning (e.g., EEG vs. EOG vs. respiratory effort). To capture this structure, `ReCoCa` first applies a channel-independent patch embedding so that each sensor's local morphology is encoded before cross-channel mixing. The resulting tokens are processed by interleaved temporal-attention and channel-attention blocks. Temporal attention uses sinusoidal RoPE (Su et al., 2024) along time, while channel attention uses a shared learnable RoPE along the sensor dimension, allowing the model to represent montage topology and stable inter-channel relationships.

*Signal Reconstruction Decoder.* Text supervision is sparse relative to the information presented in PSG. If trained only to align with captions, the encoder may discard waveform details that are not explicitly described. `ReCoCa` therefore includes a lightweight reconstruction decoder that predicts the original signal patches from the encoder latents, encour-

aging physiologically grounded representations and acting as a regularizer during multimodal pretraining.

*Modality-Conditioned Text Decoder.* PSG descriptions can target different physiological systems, and generating a single caption that covers all channels can be inefficient. We therefore group captions into four systems, $\mathcal{M} = \{\texttt{Brain}, \texttt{Respiratory}, \texttt{Cardiac}, \texttt{Somatic}\}$. During training and inference, we sample a target system $m \in \mathcal{M}$ and prepend a learnable token $[m]$ to the decoder input. This conditioning guides the decoder to focus on one system at a time, enabling *controllable* and *targeted* caption generation.

**Pretraining Objectives.** `ReCoCa` uses a composite loss function that combines *contrastive*, *reconstruction*, and *generative* objectives. Given a batch of $N$ paired examples $\{(\boldsymbol{x}_n, \boldsymbol{y}_n)\}_{n \in [N]}$, where $\boldsymbol{x}$ denotes a PSG epoch and $\boldsymbol{y}$ denotes its caption, let the sleep encoder produce a pooled embedding $\boldsymbol{s}_i$ (e.g., a CLS token) and the text encoder produce a pooled embedding $\boldsymbol{v}_i$. We first employ a symmetric InfoNCE contrastive objective (Chen et al., 2020):

$$\mathcal{L}_{\text{con}} = -\frac{1}{N} \bigg( \underbrace{\sum_{i=1}^{N} \log \frac{\exp(\text{sim}(\boldsymbol{s}_i, \boldsymbol{v}_i)/\tau)}{\sum_{j=1,\, j \neq i}^{N} \exp(\text{sim}(\boldsymbol{s}_i, \boldsymbol{v}_j)/\tau)}}_{\text{sleep-to-text}} +$$

$$\underbrace{\sum_{i=1}^{N} \log \frac{\exp(\text{sim}(\boldsymbol{v}_i, \boldsymbol{s}_i)/\tau)}{\sum_{j=1,\, j \neq i}^{N} \exp(\text{sim}(\boldsymbol{v}_i, \boldsymbol{s}_j)/\tau)}}_{\text{text-to-sleep}} \bigg),$$

where $\text{sim}(\cdot, \cdot)$ is a similarity function between embeddings, and $\tau$ denotes the temperature parameter. To further preserve signal fidelity, we reconstruct the input from the sleep encoder representations. Let $\widehat{\boldsymbol{x}}$ denote the reconstructed sleep epoch, we minimize a mean-squared error:

$$\mathcal{L}_{\text{rec}} = \frac{1}{N} \sum_{i=1}^{N} \|\boldsymbol{x}_i - \widehat{\boldsymbol{x}}_i\|_2^2.$$

*Table 1.* **Zero-shot classification and regression.** We compare `SleepLM` with fine-tuned VLMs and state-of-the-art LLMs across a broad set of tasks. Results are averaged over the SHHS (internal), MrOS (internal), and CFS (external) evaluation sets. Sleep-event IoU and balanced accuracy are averaged over {Central Apnea, Hypopnea, Oxygen Desaturation, Arousal}. Heart rate MAE is averaged over {Mean, Min, Max}, and SpO$_2$ MAE over {Min, Mean}. Channel statistics MAE is averaged over a diverse set of per-channel statistics (Appendix B.2). Detailed task setups and additional results are provided in Appendix C and D.5.

| Model | Sleep Stage | | Sleep Event | | Heart Rate | | SpO$_2$ | | Channel Stats |
|---|---|---|---|---|---|---|---|---|---|
| | AUC$^\uparrow$ | BAcc$^\uparrow$ | IoU$^\uparrow$ | BAcc$^\uparrow$ | MAE$^\downarrow$ | Recall$^\uparrow$ | MAE$^\downarrow$ | Recall$^\uparrow$ | SMAPE$^\downarrow$ |
| *Fine-tuned VLMs:* | | | | | | | | | |
| Qwen3-VL-8B-Instruct (Bai et al., 2025) | 70.2 | 51.3 | 13.5 | 59.8 | 14.25 | 22.2 | 2.73 | 18.4 | 34.62 |
| LLaVA-Next (Li et al., 2024) | 58.9 | 33.7 | 7.4 | 54.0 | 14.67 | 22.2 | 2.58 | 18.8 | 36.34 |
| *State-of-the-art LLMs:* | | | | | | | | | |
| DeepSeek-R1 (Guo et al., 2025) | 50.9 | 20.8 | 1.6 | 50.8 | 9.42 | 27.3 | **1.88** | 9.4 | - |
| Gemini 2.5 Pro (Comanici et al., 2025) | 52.2 | 24.5 | 2.8 | 51.7 | 2.01 | 10.7 | 2.20 | 12.1 | - |
| `SleepLM` | **85.4** | **76.9** | **30.4** | **74.3** | **1.97** | **35.8** | 2.24 | **39.1** | **3.15** |
| Gains | **+15.2** | **+25.6** | **+15.9** | **+14.5** | **+0.04** | **+8.5** | **-0.36** | **+20.3** | **+21.47** |

For caption generation, the multimodal text decoder conditions on the sleep embeddings and a modality token $[m]$. Using an autoregressive factorization, the captioning loss is:

$$\mathcal{L}_{\text{cap}} = -\sum_{t=1}^{T} \log \mathbb{P}_\theta(\boldsymbol{y}_t \mid \boldsymbol{y}_{<t}, \boldsymbol{x}, [m]).$$

**The `SleepLM` Family.** The final objective is a weighted combination $\mathcal{L}_{\text{ReCoCa}} = \lambda_{\text{con}} \cdot \mathcal{L}_{\text{con}} + \lambda_{\text{rec}} \cdot \mathcal{L}_{\text{rec}} + \lambda_{\text{cap}} \cdot \mathcal{L}_{\text{cap}}$, which yields a family of sleep-language models by varying $(\lambda_{\text{con}}, \lambda_{\text{rec}}, \lambda_{\text{cap}})$. For instance, setting $\lambda_{\text{rec}} = 0$ and $\lambda_{\text{cap}} = 0$ yields a CLIP-style dual encoder (Fig. 3). The full `ReCoCa` configuration uses all three losses and is our default model; we ablate these variants in Sec. 5.3.

## 5. Experiments and Results

**Datasets.** We use the pretraining splits of SHHS, MrOS, and CCSHS as the primary training corpora, and keep CFS and WSC fully held out. For zero-shot evaluation, we use the validation splits of SHHS and MrOS together with CFS to assess both internal and external generalization. Due to its higher prevalence of rare apnea subtypes, WSC is reserved for few-shot learning and unseen-concept generalization experiments. Further details are provided in Appendix B.

**Baselines.** For zero-shot and generative tasks, we compare against two classes of models. ❶ *Fine-tuned multimodal LLMs:* we adapt leading open-source vision-language models by replacing the vision encoder with our pretrained sleep encoder, then fine-tune the full system on our sleep-text corpus. We evaluate Qwen3-VL-8B-Instruct (Bai et al., 2025) and LLaVA-Next (Li et al., 2024) as representative strong open-source backbones. ❷ *Proprietary LLMs:* we evaluate strong commercial models, including Gemini 2.5 Pro (Comanici et al., 2025) and DeepSeek-R1 (Guo et al., 2025), by providing PSG as tabular time-series input with task-specific prompts. For few-shot tasks, we compare against SOTA self-supervised learning methods, MAE (He et al.,

2022) and SimCLR (Chen et al., 2020). These models are pretrained on the same PSG corpus as all other methods for a controlled comparison. Training details and prompts for zero-shot tasks are provided in Appendix A and E.2.

**Metrics.** For zero-shot classification, we report area under the ROC curve (AUROC), and balanced accuracy (BAcc). For zero-shot event localization, we report intersection-over-union (IoU). We report both symmetric mean absolute percentage error (sMAPE) and mean absolute error (MAE) for regression. For few-shot learning, we report AUROC across {5, 10, 20, 50} samples per class. Cross-modal retrieval is evaluated using Recall@1 and Recall@5 (R@K). A detailed settings by task is provided in Appendix C.

Unless otherwise stated, we report the main results using the base `ReCoCa` configuration, denoted as `SleepLM-B`. We study the effects of scaling model components in Sec. 5.2.

### 5.1. Main Results

**Zero-Shot Recognition.** We evaluate `SleepLM` in a zero-shot setting across four task categories: ❶ five-class sleep staging, ❷ sleep event localization, ❸ implicit physiological inference (HR/SpO$_2$), and ❹ explicit signal grounding (channel statistics). As shown in Table 1, `SleepLM` consistently outperforms both proprietary LLMs and fine-tuned VLMs across all zero-shot tasks. The baselines exhibit distinct failure modes. Proprietary LLMs, even when given tabular PSG descriptors, produce near-random predictions for sleep stages and event identification, yet perform relatively well on HR and SpO$_2$ regression. This suggests that strong LLMs can extract and manipulate explicit numeric summaries, but struggle to map low-level signal descriptors into higher-level sleep and event states. In contrast, fine-tuned VLMs are consistently suboptimal, suggesting that simply swapping in a sleep encoder does not yield effective multimodal fusion for dense physiological time series.

*Table 2.* **Zero-shot cross-modal retrieval.** We evaluate text-to-signal (**top**) and signal-to-text (**bottom**) retrieval for `SleepLM` and LLM baselines. "–" indicates that the task is infeasible for an LLM baseline due to context limits. Full results are in Appendix D.6.

| Model | n=100 | | n=2000 | |
|---|---|---|---|---|
| | R@1↑ | R@5↑ | R@1↑ | R@5↑ |
| *Text → Signal Retrieval* | | | | |
| DeepSeek-R1 (Guo et al., 2025) | - | - | - | - |
| Gemini 2.5 Pro (Comanici et al., 2025) | - | - | - | - |
| `SleepLM` | **94.3** | **99.0** | **82.5** | **91.9** |
| *Signal → Text Retrieval* | | | | |
| DeepSeek-R1 (Guo et al., 2025) | 1.7 | 5.7 | - | - |
| Gemini 2.5 Pro (Comanici et al., 2025) | 4.0 | 10.7 | - | - |
| `SleepLM` | **94.3** | **99.7** | **80.9** | **91.5** |

*Table 3.* **Zero-shot generalization to unseen concepts.** We report performance on two held-out respiratory event classification tasks, where `SleepLM` remains robust across both settings.

| Model | Mixed Apnea | | Obstructive Apnea | |
|---|---|---|---|---|
| | F1↑ | BAcc↑ | F1↑ | BAcc↑ |
| Gemini 2.5 Pro | 33.4 | 48.3 | 15.6 | 51.4 |
| `SleepLM` | **78.2** | **81.8** | **79.7** | **77.1** |

**Zero-Shot Cross Modal Retrieval.** We evaluate cross-modal alignment via retrieval in both directions (signal-to-text and text-to-signal), reporting R@1 and R@5. As shown in Table 2, `SleepLM` substantially outperforms LLM baselines, which are only slightly above random in this dense retrieval setting. On a 100-sample validation subset, `SleepLM` achieves near-perfect accuracy. On the full 2,000-sample validation set, where LLM baselines are not feasible due to context-length constraints, `SleepLM` still maintains strong retrieval performance. These results confirm that our pretraining objective learns a precise mapping between physiological states and their language descriptions. Later in Sec. 5.2, we further demonstrate that `SleepLM` learns a continuous embedding space that supports practical retrieval beyond exact matches, enabling the search of semantically related data clusters.

**Generalization to Unseen Concepts.** A key property of foundation models is the ability to generalize to concepts that are not explicitly observed during training. We evaluate `SleepLM` on two held-out clinical events: "Mixed Apnea" and "Obstructive Apnea". These labels are absent from the training vocabulary, requiring the model to extrapolate from related physiological patterns (e.g., "Central Apnea"). As Table 3 confirms, while LLM baselines perform at chance, `SleepLM` achieves approximately 80% F1 and accuracy on these unseen events. To probe the source of this behavior, we visualize the embedding space with UMAP (Fig. 4). We observe that `SleepLM` places mixed and "Obstructive Apnea"

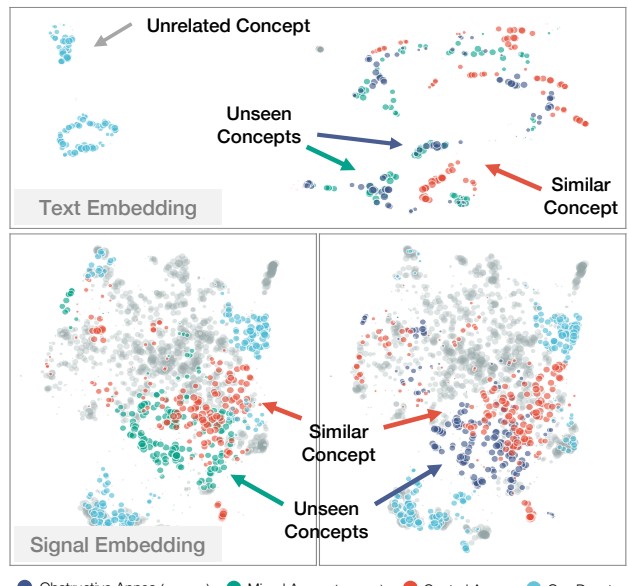

● Obstructive Apnea (**unseen**)  ● Mixed Apnea (**unseen**)  ● Central Apnea  ● Oxy Desat.

*Figure 4.* **Zero-shot generalization analysis of `SleepLM`.** We visualize text and signal embeddings with UMAP as a case study of zero-shot concept transfer. `SleepLM` is capable of clustering previously unseen concepts to semantically related seen concepts.

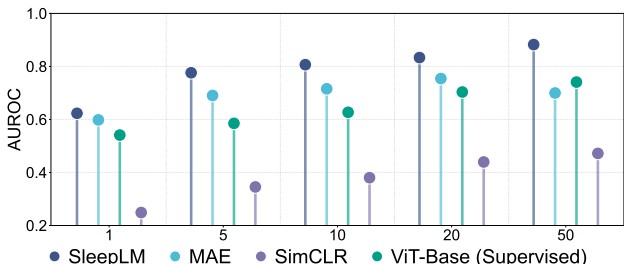

● SleepLM  ● MAE  ● SimCLR  ● ViT-Base (Supervised)

*Figure 5.* **Few-shot adaptation to downstream tasks.** We compare `SleepLM` with SSL and supervised baselines under varying numbers of labeled samples per class. `SleepLM` shows higher data efficiency and stronger performance across all shot regimes.

near "Central Apnea" (a seen concept), while separating them from physiologically distinct events such as "Oxygen Desaturation". This structure suggests that `SleepLM` learns a latent concept that spans both seen and unseen variants in the aligned text and physiology spaces.

**Few-Shot Learning.** We evaluate the transferability of `SleepLM` by isolating the sleep encoder and comparing it with SOTA SSL baselines and a supervised ViT model (Dosovitskiy et al., 2021). We freeze encoder backbones and train a linear probe using a small labeled samples per class (i.e., $\{1, 5, 10, 20, 50\}$) on the held-out WSC cohort. As shown in Fig. 5, `SleepLM` consistently outperforms both SSL and supervised baselines on sleep staging. With only 50 samples per class, `SleepLM` reaches approximately 0.90 AUC, indicating strong data efficiency. Beyond general pretraining recipes, we further validate our approach by comparing `SleepLM` against specialized domain architectures.

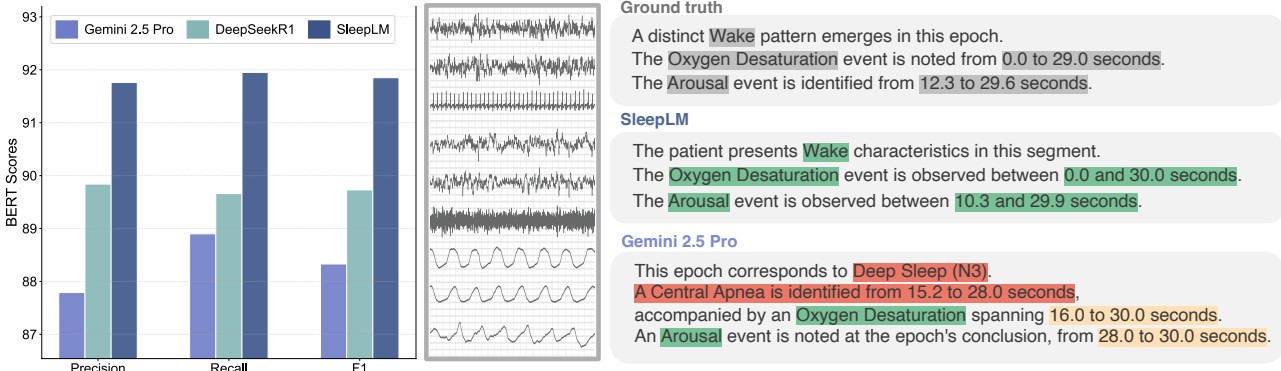

*Figure 6.* **Sleep caption generation.** `SleepLM` captures both high-level sleep stages and fine-grained localized events, while LLM baselines often fail to recognize or localize. Additional examples, including fine-tuned VLM outputs, are provided in Appendix D.9. The generation scores are provided in Table 20.

We conduct few-shot linear probing runs and full finetuning runs against diverse types of domain specific models: (1) *Supervised sleep stage models* (Guillot & Thorey, 2021; Perslev et al., 2021), (2) *Sleep FMs* (Thapa et al., 2024), and (3) *General time series FMs* (Ansari et al., 2024). Specifically, to ensure a fair comparison, the domain-specific models are trained on the same data splits as `SleepLM`. We then isolated `SleepLM`'s sleep encoder and evaluate all models via linear probing and full fine-tuning on the completely unseen CFS dataset. Notably, for Chronos-2, we fine-tune its public checkpoint to properly assess its capacity as a general time-series foundation model. Ultimately, `SleepLM` demonstrates highly competitive performance across all tasks and settings. Together, these findings indicate that the semantic structure learned through caption-based supervision produces more discriminative and transferable features than standard reconstruction or invariance objectives, consistently matching or exceeding prior domain-specific state-of-the-art methods. Detailed results are provided in Appendix D.4.

### 5.2. Analyses

**Sleep Caption Generation.** We assess the generation quality of `SleepLM` by comparing its captions with ground-truth text and with captions from Gemini 2.5 Pro. As confirmed in Fig. 6, `SleepLM` produces concise, clinically accurate descriptions that capture both sleep stage and the timing of localized events. In contrast, Gemini 2.5 Pro frequently introduces incorrect associations and fails to reflect the underlying signal morphology.

**Localization Sensitivity.** One key capability is whether the model captures *when* an event occurs, rather than only its presence. To test this, we run a controlled perturbation study (Fig. 7). We select an epoch containing a ground-truth event (e.g., hypopnea) and construct synthetic captions that are identical except for their timestamp intervals. We then compute the cosine similarity between the fixed signal embedding and the text embeddings of these temporally shifted

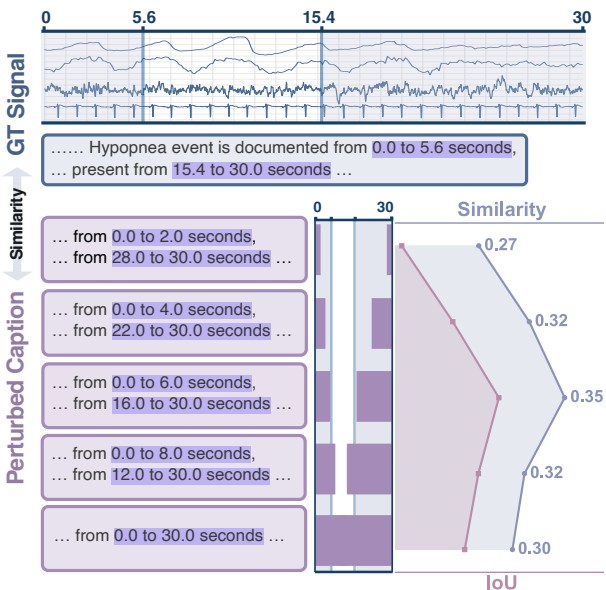

*Figure 7.* **Localization sensitivity of** `SleepLM`**.** Given a fixed signal embedding and its caption, we progressively shift the event timestamp in the caption and compare the resulting text embeddings to the signal embedding. Embedding similarity increases with the IoU between the ground-truth and perturbed timestamps, peaking near the correct alignment.

captions. The similarity shows a strong linear relationship with timestamp IoU: it peaks near the correct alignment and decreases as overlap diminishes. This indicates that `SleepLM` learns temporally grounded representations that are sensitive to fine-grained localization in a zero-shot setting. Additional examples are provided in Appendix E.5.

**Embedding Space Continuity.** To further visualize the structure of the learned latent space, we analyze retrieval results for a single PSG epoch. Fig. 8 shows the top three retrieved captions from the pool, ranked by decreasing similarity. For clarity, we display only the global sleep stage and event descriptions. We observe a smooth semantic gradient: higher similarity scores consistently return captions

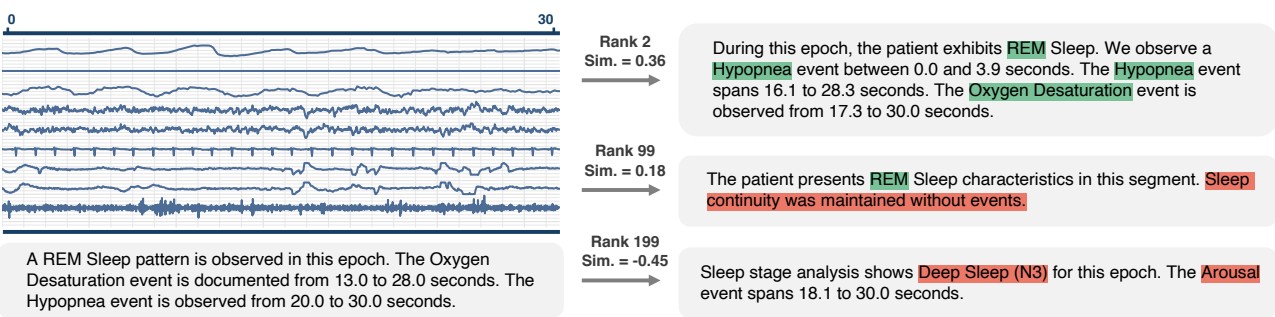

*Figure 8.* **Semantic retrieval continuity of** SleepLM. Retrieval results show a smooth semantic gradient: high-similarity captions (green) match the query's physiological state, while lower-scoring results (red) correspond to distinct conditions. This indicates that embedding distance of SleepLM reflects physiological similarity. We provide more examples in Appendix E.4.

that match the query physiology, while lower scores correspond to physiologically distinct states. This indicates that SleepLM learns a continuous and semantically meaningful manifold in which embedding distance reflects physiological similarity, enabling data exploration based on semantic proximity rather than exact concept matches. Additional examples are provided in Appendix E.4.

**Scaling Behavior (Appendix D.1 & D.2):** We study how SleepLM scales with both model size and data diversity. For model scaling, we train three variants, SleepLM-T (38M), SleepLM-S (180M), and SleepLM-B (410M). As shown in Table 10, performance improves consistently as parameter count increases, with particularly clear gains in channel-statistics regression and cross-modal retrieval, and no evidence of saturation at the base scale. Model specifications are summarized in Table 11. For data scaling, we compare single-source pretraining (SHHS only) with multi-source pretraining (SHHS + MrOS + CCSHS). Multi-source training improves all metrics and even outperforms the single-source baseline on the internal SHHS evaluation set, indicating that our captioning pipeline provides stable supervision across cohorts and that added source diversity strengthens, rather than harms, generalization.

**Full-Night Reporting (Appendix D.8).** To connect with clinical practice, we aggregate epoch-level predictions across full-night recordings to derive standard diagnostic metrics, including the apnea–hypopnea index (AHI) and wake after sleep onset (WASO). We randomly select five SHHS subjects and run SleepLM in a sliding-window manner over each night, then summarize the fine-grained outputs into full-night statistics (details in Appendix D.8). SleepLM shows strong concordance with manual scoring and remains stable across thousands of epochs, whereas fine-tuned VLM baselines exhibit drift over long sequences. Fig. 19 shows an example report constructed from these derived statistics, illustrating that SleepLM can translate longitudinal PSG into actionable, physician-oriented summaries.

*Table 4.* SleepLM **variant ablations on representative tasks.** The full ReCoCa configuration consistently performs best, highlighting the benefit of our sleep-specific design choices. Complete results are provided in Appendix D.

| Arch | Stage Acc$^\uparrow$ | Event IoU$^\uparrow$ | T2S R@1$^\uparrow$ | sMAPE$^\downarrow$ |
|---|---|---|---|---|
| Cap (Tschannen et al., 2023) | 72.0 | 29.2 | - | 7.20 |
| CLIP (Radford et al., 2021) | 74.6 | - | 74.3 | - |
| CoCa (Yu et al., 2022) | 76.0 | 27.4 | 77.5 | 7.34 |
| ReCoCa | **76.9** | **30.4** | **82.5** | **3.15** |

### 5.3. Ablation Studies

**ReCoCa vs. other** SleepLM **variants.** We compare ReCoCa with other pretraining formulations that can be instantiated within the SleepLM framework. As described in Sec. 4, we obtain CLIP-, Cap-, and CoCa-style variants by toggling the contrastive ($\mathcal{L}_{con}$) and captioning and captioning ($\mathcal{L}_{cap}$) objectives. To ablate our channel-specific sleep encoder, we also replace it with a standard encoder backbone consisting of convolutional feature extraction followed by a temporal transformer (Carter & Tarassenko, 2024). All models are matched to roughly the same parameter size. Comparing these variants in Table 4 reports representative metrics across tasks and shows that ReCoCa consistently performs best across most classification, regression, and event-localization evaluations, highlighting the benefit of our sleep-specific architectural choices.

**Sleep reconstruction as regularization.** We hypothesize that the mismatch between dense PSG and sparse text can cause the encoder to discard fine-grained morphology under text-only supervision. To test this, we ablate the reconstruction objective by setting $\mathcal{L}_{rec} = 0$. As shown in Table 5, removing reconstruction consistently degrades performance on discriminative tasks, supporting the role of reconstruction as a regularizer that preserves physiologically meaningful details. More details are provided in Appendix D.3.

**Multilevel caption supervision.** An important design of our captioning pipeline is the integration of both low-level grounding (channel/local) and high-level summaries (global). To assess the value of low-level supervision,

*Table 5.* **Ablation results on sleep reconstruction:** We show results of SleepLM trained with and without sleep reconstruction. **Ablation results on multilevel captions:** We compare the results between SleepLM trained with and without low level channel caption. All results are shown on the high-level sleep stage classification and sleep event identification tasks and averaged across all datasets.

| Setting | Sleep Stage | | Sleep Event | |
|---|---|---|---|---|
| | AUC$^\uparrow$ | Acc$^\uparrow$ | IoU$^\uparrow$ | Acc$^\uparrow$ |
| *Sleep reconstruction* | | | | |
| w/o recon. loss | 83.9 | 74.1 | 29.6 | 73.2 |
| with recon. loss | **85.4** | **76.9** | **30.4** | **74.3** |
| *Channel caption* | | | | |
| w/o channel caption | 84.6 | 75.6 | 30.2 | 73.7 |
| with channel caption | **85.4** | **76.9** | **30.4** | **74.3** |

we compare the full-caption model (channel + local + global) with an ablation trained only on global captions. Results in Table 5 demonstrate that the model trained with multilevel supervision consistently outperforms other baselines, including on high-level tasks such as sleep staging and event classification. This suggests that learning grounded waveform descriptors strengthens the representations used to infer broader physiological states. More details are provided in Appendix D.3.

## 6. Discussion

**Limitations.** While promising, SleepLM is a research prototype and is not clinically validated for diagnosis, treatment, or medical decision-making. In addition, our study focuses on five PSG cohorts curated from NSRR; further work may extend the data coverage to assess robustness to broader clinical variability, devices, and patient populations.

**Conclusion.** We present SleepLM, the first family of sleep-language foundation models that unlock human sleep understanding through natural language. By curating the first large-scale sleep-text dataset and designing a unified pretraining objective ReCoCa, we support joint learning of language and physiological time-series at scale. We verify that SleepLM achieves strong performance across diverse tasks while enabling new capabilities such as language-guided localization and generalization to unseen concepts.

## Impact Statement

This paper advances machine learning methods for understanding and describing sleep physiology from polysomnography (PSG). By aligning multi-channel PSG with natural language, SleepLM supports more accessible and interpretable sleep analysis, including patient-facing explanations, clinician-oriented summaries, and scalable research on sleep as a longitudinal health signal. If validated and in-

tegrated responsibly, such tools may reduce manual scoring burden, improve triage, and help broaden access to sleep medicine in settings with limited specialist capacity.

At the same time, sleep-language models may raise safety and ethics considerations. Generated text may be incorrect or overly confident, and performance can vary across cohorts, devices, and populations due to dataset shift and bias. PSG data are sensitive, and misuse or leakage could harm privacy. For these reasons, SleepLM should be treated as an assistive research system rather than an autonomous diagnostic tool, and any clinical use would require careful validation, monitoring, and adherence to privacy and regulatory requirements.

## Acknowledgments

We gratefully acknowledge the support by Amazon Science Hub and UCLA DataX. Any opinions, findings, conclusions, or recommendations expressed in this material are those of the author(s) and do not necessarily reflect the views of the funders.

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

## A. Training Details

**Training `ReCoCa`:** We train `ReCoCa` using the AdamW optimizer with a learning rate of 1e-4 and a cosine annealing schedule. The training runs for 15 epochs with a 5,000-step linear warmup. We utilize a global batch size of 384 distributed across 4 NVIDIA H100 GPUs. The loss components are weighted as follows: $\lambda_{con} = 1.0$, $\lambda_{cap} = 2.0$, and $\lambda_{rec} = 0.1$. Gradients are clipped at a norm of 1.0. Training typically converges within approximately 48 hours.

**Baseline Finetuning Strategy:** For the multimodal LLM baselines (Qwen3-VL and LLaVA-NeXT), we adopt the two-stage modality adaptation protocol proposed in LLaVA-Next (Li et al., 2024).

- **Stage 1 (Alignment):** We freeze both the pretrained Sleep Encoder (initialized from `ReCoCa`) and the LLM backbone, training only the projector and token pooler layers. This stage aligns the sleep feature space with the LLM's embedding space.

- **Stage 2 (Finetuning):** We unfreeze the projector, pooler, and Sleep Encoder, and apply Low-Rank Adaptation (LoRA) to the LLM backbone.

Due to computational constraints, we use LoRA with rank $r = 16$, $\alpha = 32$, and dropout $p = 0.05$. We use a batch size of 8 with gradient accumulation every 16 steps. The models are warmed up for 500 steps followed by one full epoch of finetuning. These baseline experiments require significantly higher compute, taking 96–144 hours on the same 4×H100 hardware setup.

## B. Dataset Details

### B.1. Data Split & Preprocessing

To evaluate generalization and zero-shot performance, we implement a strict splitting strategy. We partition SHHS and MrOS into pretraining and internal evaluation sets on a subject level. We utilize CCSHS exclusively for training. Crucially, we hold out CFS and WSC entirely to serve as external validation datasets. For evaluation, we sample a fixed subset of 2,000 epochs from the validation partition of each dataset. This sampling strategy is necessary to accommodate the prohibitive computational and financial costs associated with performing inference on the full validation set using proprietary LLMs. To preserve statistical validity, this subset is strictly stratified at the subject level, ensuring maximal biological variance despite the reduced sample size. Additionally, to ensure high representativeness, we strictly stratified these validation samples across 930 distinct patients (SHHS: 409, MrOS: 193, CFS: 328), capturing a truly diverse population. The complete split information is available in Table 6, and we provide the demographic distribution over different cohorts used in Table 7.

*Table 6.* Distribution of number of epochs across different datasets and splits

| Split | Internal Datasets | | | External Datasets | | Total |
|---|---|---|---|---|---|---|
| | SHHS | MROS | CCSHS | CFS | WSC | |
| Train | 6,985,633 | 3,569,317 | 552,693 | 0 | 0 | 11,107,643 |
| Valid | 2,000 | 2,000 | 0 | 2,000 | 2,000 | 8,000 |
| Total | 6,987,633 | 3,571,317 | 552,693 | 2,000 | 2,000 | 11,115,643 |

Missing channels are zero-padded to maintain consistency. Given the heterogeneity of the source data (varying sampling rates, device ranges), we unify all signals to a fixed sampling rate of 64Hz. We apply manual quality control over all night's data to trim excessive wakefulness or non-wear period at the start and end of recordings to reduce extreme sensor noise. Finally, we apply z-score normalization on a per-night basis. For respiratory channels specifically, we apply area-dependent z-score normalization to ensure consistent amplitude scaling over time.

### B.2. Statistics Details

For our Channel Captions, we defined a variable set of statistics for each channel in order to provide diverse, fine-grained, clinically relevant information. The specific statistics and their relevant channels are provided in Table 6.

*Table 7.* **Age distribution across cohorts in train and validation splits.** Values are percentages. This demographic skew is expected and aligns with real world clinical baselines, as the prevalence of sleep disorders dramatically increases with advanced age.

| | Train | | | | | Validation | | | |
|---|---|---|---|---|---|---|---|---|---|
| **Age** | **SHHS** | **MROS** | **CCSHS** | **All** | **Age** | **SHHS** | **MROS** | **CFS** | **All** |
| 0–18 | 0.0 | 0.0 | 78.1 | 3.3 | 0–18 | 0.0 | 0.0 | 20.7 | 7.3 |
| 18–35 | 0.0 | 0.0 | 21.9 | 0.9 | 18–35 | 0.0 | 0.0 | 21.0 | 7.4 |
| 35–50 | 12.9 | 0.0 | 0.0 | 8.4 | 35–50 | 11.0 | 0.0 | 26.5 | 14.2 |
| 50+ | 87.1 | 100.0 | 0.0 | 87.3 | 50+ | 89.0 | 100.0 | 31.7 | 71.1 |

*Table 8.* **Channel Statistics Configuration:** a detailed list of what statistics we compute for each channel and included into the channel caption

| Channel | Statistic | Description |
|---|---|---|
| ECG | hr_mean | estimated beat rate |
| | rmssd_30s | ultra-short HRV (RMSSD) |
| HR | min | minimum value |
| | max | maximum value |
| | mean | mean |
| SPO2 | min | minimum value |
| | mean | mean |
| ABD | rr_bpm | respiratory rate |
| | rr_iqr | breath interval variability |
| THX | rr_bpm | respiratory rate |
| | rr_iqr | breath interval variability |
| AF | rr_bpm_af | airflow rate |
| | flow_flatness | inspiratory flow flatness |
| EOG_E1_A2 | sem_power | slow eye movement relative power |
| | rem_power | REM saccadic relative power |
| EOG_E2_A1 | sem_power | slow eye movement relative power |
| | rem_power | REM saccadic relative power |
| EMG_Chin | mdf_hz_10_30 | median frequency |
| | tail_ratio | burst intensity ratio |
| | env_centroid_hz | burst modulation rate |
| EEG_C3_A2 | delta_power | delta relative power |
| | theta_power | theta relative power |
| | alpha_power | alpha relative power |
| | beta_power | beta relative power |
| EEG_C4_A1 | delta_power | delta relative power |
| | theta_power | theta relative power |
| | alpha_power | alpha relative power |
| | beta_power | beta relative power |
| POS | - | - |

# C. Tasks Setup

## C.1. Zero-Shot Task Definitions

We assess the model's zero-shot capabilities across the following four categories.

**1. Sleep Staging Classification:** We perform standard 5-class classification (Wake, N1, N2, N3, REM). To classify an epoch, we calculate the cosine similarity between the signal embedding $z_{cls}$ and the average text embedding of a diverse set of template captions (e.g., "The patient is in N2 sleep") for each stage. We report the AUC and balanced accuracy. The full list of template prompts is provided in Appendix E.3.

**2. Sleep Event Localization & Classification:** vWe evaluate the ability to identify and localize specific clinical events: arousal, central apnea, hypopnea, and oxygen desaturation. Note that obstructive apnea and mixed apnea are explicitly held out from this evaluation to test generalization on unseen concept.

- **Classification:** We treat this as a binary classification task per event type and report balanced accuracy.

- **Localization:** We parse the start and end timestamps from the generated caption and compare them against ground truth annotations using IoU.

**3. Implicit Physiological Inference:** As described in the method section, Heart Rate and SpO2 signals are excluded from the encoder input. We evaluate the model's ability to infer these vitals purely from cross-channel correlations in the remaining sensors.

- **Statistics:** We parse the predicted statistics (mean, min, max) from the generated caption and calculate the MAE against the ground truth.

- **Trends:** We extract predicted trend windows. Due to the subjective definition of "trends" in physiological signals, we report Recall rather than IoU to avoid penalizing valid detections that slightly differ from heuristic ground truth boundaries.

**4. Explicit Signal Grounding (Channel Statistics):** To assess the model's physical understanding of the visible signals provided to the encoder, we evaluate its ability to estimate clinically relevant signal statistics (e.g., EEG variance, EMG power). We parse the numeric values from the generated channel-specific captions and report the Symmetric Mean Absolute Percentage Error (sMAPE).

## C.2. Unseen Concept Classification

To rigorously test generalization, we curate a balanced binary classification dataset from the external WSC dataset, which was held out entirely during pretraining. The dataset consists of 500 positive and 500 negative samples for each of the two unseen events (mixed apnea and obstructive apnea).

Zero-Shot Prototype Construction: Since the specific labels for these apneas were not present in the pretraining vocabulary, we utilize a zero-shot prototype approach by computing the cosine similarity between the signal embedding and two text anchors:

1. **Positive Anchor:** The average embedding of a diverse set of "event presence" templates (e.g., "An obstructive apnea event").

2. **Negative Anchor:** Constructing a negative anchor is non-trivial, as our pretraining corpus lacks explicit negation concepts (e.g., "No obstructive apnea"). We therefore utilize a global "No event at all" anchor as a substitute.

Note on Anchor Noise: It is important to note that the negative anchor described above is inherently noisy. A "negative" sample in this binary task is defined as "not obstructive Aapnea," but the epoch may still contain other events (e.g., a hypopnea or arousal). By using "No event at all" as the negative anchor, we inadvertently penalize the model if it detects these other valid events. The high performance reported in the main text is achieved despite this structural disadvantage, underscoring the model's discriminative precision.

## C.3. Few-Shot Evaluation

To assess representation quality in data-scarce regimes, we conduct a linear probing evaluation on the external WSC dataset using the following setup.

Baselines: We compare `ReCoCa` against four distinct baselines, all controlled to have approximately the same parameter count:

- **SSL Baselines:** MAE (Masked Autoencoder) and SimCLR (Contrastive Learning), trained on the same internal corpus as `ReCoCa` to ensure fair comparison of pretraining objectives.
- **Supervised Baselines:** WideResNet (Zagoruyko & Komodakis, 2016) and ViT (Dosovitskiy et al., 2021), trained from scratch.

Protocol: We simulate data scarcity by providing strictly $K$ labeled samples per class, where $K \in \{1, 5, 10, 20, 50\}$. Frozen Encoder: The backbone weights of all models are frozen to evaluate the quality of the static pretrained representations. Linear Probe: A simple linear classifier is trained on top of the fixed embeddings using the limited $K$-shot training set. Tasks: Evaluation is performed on two downstream tasks: 5-class sleep stage classification and binary oxygen desaturation detection.

# D. Additional Results

This section presents supplementary experimental results that substantiate the claims in the main manuscript and include the full, unaggregated results underlying the summarized evaluations reported in the main text.

## D.1. Scaling on Dataset

*Table 9.* **Effect of pretraining data scale on downstream performance** (internal: SHHS; external: CFS). Best is **bold**; second best is underlined.

| Internal | | | | | External | | | | |
|---|---|---|---|---|---|---|---|---|---|
| *Pretraining data* | Stage Acc$^\uparrow$ | Event IoU$^\uparrow$ | TtS R@1$^\uparrow$ | SMAPE$^\downarrow$ | *Pretraining data* | Stage Acc$^\uparrow$ | Event IoU$^\uparrow$ | TtS R@1$^\uparrow$ | SMAPE$^\downarrow$ |
| SHHS only | 77.0 | 31.2 | 96.0 | 3.80 | SHHS only | 70.5 | 27.2 | 73.5 | 5.38 |
| Multi-source | **79.6** | **31.9** | **96.1** | **2.73** | Multi-source | **74.2** | **28.2** | **78.7** | **3.99** |

In Table 9, we present the results of training on single vs multisource datasets. Our results demonstrate that multisource data pretraining not only helps the performance on the external datasets, which is expected, but also brings a significant performance on the internal dataset. This shows that our captioning pipeline is robust and extends beyond datasets' boundary.

## D.2. Scaling on Model size

*Table 10.* **Effect of `ReCoCa` parameter size on downstream performance** (internal: (SHHS+MROS); external: CFS). Best is **bold**; second best is underlined.

| Internal | | | | | External | | | | |
|---|---|---|---|---|---|---|---|---|---|
| *Pretraining data* | Stage Acc$^\uparrow$ | Event IoU$^\uparrow$ | TtS R@1$^\uparrow$ | SMAPE$^\downarrow$ | *Pretraining data* | Stage Acc$^\uparrow$ | Event IoU$^\uparrow$ | TtS R@1$^\uparrow$ | SMAPE$^\downarrow$ |
| ReCoCa-Tiny | 75.2 | 25.7 | 63.1 | 8.23 | SleepLM-T | 71.9 | 27.2 | 52.5 | 8.98 |
| ReCoCa-Small | 77.1 | 29.4 | 76.9 | 5.24 | SleepLM-S | 72.8 | 27.2 | 65.4 | 6.38 |
| ReCoCa-Base | **78.3** | **31.3** | **84.4** | **3.13** | SleepLM-B | **74.2** | **28.2** | **78.7** | **3.99** |

To assess the scalability of the framework, we evaluate three variants of `SleepLM` with increasing capacity: `SleepLM-T` (38M), `SleepLM-S` (180M), and `SleepLM-B` (410M). Detailed architectural specifications for each configuration are provided in Table 11.

As presented in Table 10, we observe consistent monotonic improvements in performance as parameter count increases. This trend is particularly pronounced in tasks requiring fine-grained semantic understanding, such as cross-modal retrieval

*Table 11.* **Architecture specs for ReCoCa variants.**

| Model | Sleep encoder | | | Text encoder | | | Sleep decoder | | | #Params |
|---|---|---|---|---|---|---|---|---|---|---|
| | Head | Layer | Dim | Head | Layer | Dim | Head | Layer | Dim | |
| SleepLM-T | 8 | 2 | 256 | 8 | 3 | 256 | 8 | 3 | 256 | 38M |
| SleepLM-S | 12 | 2 | 768 | 12 | 4 | 768 | 12 | 4 | 768 | 180M |
| SleepLM-B | 12 | 6 | 768 | 12 | 12 | 768 | 12 | 12 | 768 | 410M |

and channel statistics regression. Notably, performance does not saturate at the Base scale (410M), suggesting that the `SleepLM` architecture is capable of effectively leveraging larger parameter budgets for further gains.

### D.3. Ablation Study

We conduct two ablation studies to validate our design choices, with full numerical comparisons presented in the Table 5.

**Sleep Reconstruction as Regularization:** We posit that reliance on sparse text supervision alone creates an "information density gap" when paired with dense physiological signals, potentially leading the encoder to discard nuanced waveform features which leads to feature collapse. By enforcing a reconstruction objective ($\mathcal{L}_{rec}$), we compel the model to retain a complete representation of the input signal, ensuring that fine-grained morphological details are preserved alongside semantic abstractions. This acts as a critical regularizer, grounding the latent space in the physical reality of the signal rather than just the linguistic approximation. As shown in the results, removing this objective consistently degrades performance across all discriminative tasks.

**Multilevel Supervision of Captions:** Our data pipeline integrates low-level channel captions alongside high-level global summaries to foster a "bottom-up" understanding of sleep physiology. The intuition is that robust high-level inference (e.g., sleep staging) relies on the accurate detection of fundamental waveform statistics and local morphologies. By explicitly supervising the model on these granular details, we prevent it from overfitting to abstract labels and instead encourage a hierarchical learning process. The results confirm that this low-level grounding provides an improvement in performance even on global classification tasks compared to a high-level-only baseline.

### D.4. Fewshot Results

*Table 12.* **Few-shot classification results on test set** (AUROC, %). Best is **bold**; second best is underlined.

| Model | # shots | | | | |
|---|---|---|---|---|---|
| | 1 | 5 | 10 | 20 | 50 |
| *Oxygen Desaturation* | | | | | |
| Wide-ResNet-50 | 53.9 | 51.6 | 55.1 | 54.3 | 61.5 |
| ViT-Base | 55.5 | 57.6 | 51.4 | 58.3 | 56.6 |
| MAE | 54.0 | **60.8** | 58.9 | 57.4 | 59.0 |
| SimCLR | 52.2 | 57.7 | 56.1 | 61.7 | 63.4 |
| ReCoCa | **56.4** | 59.3 | **59.6** | **65.0** | **65.2** |
| *Sleep Stage (5-class, macro-OvR)* | | | | | |
| Wide-ResNet-50 | 46.4 | 52.1 | 64.8 | 59.1 | 79.1 |
| ViT-Base | 54.1 | 58.5 | 62.7 | 70.4 | 74.1 |
| MAE | 59.8 | 69.0 | 71.6 | 75.4 | 70.0 |
| SimCLR | 24.9 | 34.6 | 38.0 | 44.0 | 47.2 |
| ReCoCa | **62.3** | **77.6** | **80.7** | **83.3** | **88.3** |

In this section, we provide a comprehensive breakdown of the few-shot transfer learning experiments introduced in the main text. To assess the quality of the learned representations in data-scarce regimes, we isolate the sleep encoder of `SleepLM` and compare it against state-of-the-art SSL baselines (MAE (He et al., 2022), SimCLR (Chen et al., 2020)) and supervised architectures (ViT (Dosovitskiy et al., 2021)) on the held-out WSC cohort. We freeze the encoder bodies of all models and train a linear probe using strictly $K$ labeled samples per class, where $K \in \{1, 5, 10, 20, 50\}$. This setup explicitly tests the discriminative power of the static, pretrained features without the benefit of fine-tuning. While the main text highlights performance on sleep staging, we present here the extended evaluation covering additional oxygen desaturation detection

task. As detailed in the Table 18, `SleepLM` consistently outperforms specialized SSL and supervised baselines across these diverse physiological targets, confirming that the semantic structure imposed by our captioning objective yields features that are significantly more robust and transferable than those learned via standard reconstruction or invariance-based objectives.

*Table 13.* **Full finutuning AUC for `SleepLM` against domain specific models.**

| Model | Sleep Stage | Arousal | Hypopnea | Oxy. Desat. |
|---|---|---|---|---|
| Chronos-2 | **87.6** | 87.9 | **85.0** | 83.7 |
| SleepFM | 85.4 | 86.7 | 81.7 | 80.3 |
| RobustSleepNet | 85.8 | – | – | – |
| U-Sleep | 83.8 | – | – | – |
| `SleepLM` | 87.2 | **89.2** | 83.0 | **84.1** |

*Table 14.* **Few shot linear probing AUC under 10-shot and 50-shot settings for `SleepLM` against domain specific models.**

| Model | Sleep Stage | | Arousal | | Hypopnea | | Oxyen Desaturation | |
|---|---|---|---|---|---|---|---|---|
| | 10 | 50 | 10 | 50 | 10 | 50 | 10 | 50 |
| Chronos-2 | 59.6 | 60.4 | 64.6 | 68.3 | 50.1 | 65.1 | 60.0 | 66.8 |
| SleepFM | 64.9 | 74.7 | **70.2** | 76.8 | **60.4** | 74.0 | 61.8 | 72.8 |
| RobustSleepNet | 69.1 | 76.4 | – | – | – | – | – | – |
| U-Sleep | 61.8 | 68.9 | – | – | – | – | – | – |
| `SleepLM` | **71.1** | **79.9** | 66.5 | **79.5** | 59.3 | **75.1** | **62.3** | **76.8** |

In addition to general pretraining recipe, we further conduct few-shot linear probing runs and full finetuning against diverse types of domain specific models: (1) Supervised sleep stage models: RobustSleepNet (Guillot & Thorey, 2021), U-Sleep (Perslev et al., 2021), (2) Sleep FMs: SleepFM (Thapa et al., 2024), and (3) General time series FMs: Chronos2 (Ansari et al., 2025). When necessary, models are retrained using the exact same settings in Appendix A and B for direct comparisons on the external CFS dataset.

To compare against domain-specific models on their native tasks, while these specialized models are designed for a 'pretrain-then-finetune' paradigm, we evaluated SleepLM under the exact conditions for the fairest comparison. Specifically, we pretrained these models using the same splits as SleepLM. We then isolated SleepLM's sleep encoder and perform few-shot linear probing (Table 14) and fully finetuning (Table 13) over all models on the training split of the unseen CFS dataset. Final AUCs are reported on the full test split. For Chronos-2, we finetuned its public pretrained checkpoint to appropriately assess it as a general time-series FM.

Overall, SleepLM remains competitive across tasks and settings. Also, generally FMs perform better than smaller specialized models. We attribute this in part to the substantially larger parameter scale of FMs relative to these specialized baselines. This likely stems from two factors: (1) the established scaling behavior with model size that we also observe for SleepLM (Table 10, Appendix D.2), and (2) the ability of larger models to learn representations that transfer better across tasks and datasets. This is further supported by the strong performance of Chronos-2 in our full fine-tuning experiments.

### D.5. Classification Raw Results

In this section, we present the granular, unaggregated results for sleep stage classification and sleep event localization & classification. We benchmark `ReCoCa` against a comprehensive suite of baselines, categorized into three groups:

1. **Proprietary LLMs:** Leading general-purpose models (Gemini 2.5 Pro (Comanici et al., 2025), DeepSeek-R1 (Guo et al., 2025)).

2. **Finetuned VLMs:** Open-source vision-language models adapted for sleep (Qwen3-VL-8B-Instruct (Bai et al., 2025), LLaVA-Next (Li et al., 2024)).

3. **Standard Architectures:** `SleepLM` variants utilizing standard multimodal objectives (CLIP (Radford et al., 2021), Cap (Tschannen et al., 2023), CoCa (Yu et al., 2022)).

*Table 15.* **Zero-shot sleep stage and event detection results across datasets.** Best is **bold**; second best is underlined.

| Model | Sleep Stage | | | Central Apnea | | | Hypopnea | | | Oxygen Desaturation | | | Arousal | | |
|---|---|---|---|---|---|---|---|---|---|---|---|---|---|---|---|
| | F1↑ | AUC↑ | BAcc↑ | IoU↑ | F1↑ | BAcc↑ | IoU↑ | F1↑ | BAcc↑ | IoU↑ | F1↑ | BAcc↑ | IoU↑ | F1↑ | BAcc↑ |
| *CFS:* | | | | | | | | | | | | | | | |
| Qwen3-VL-8B-Instruct | 46.9 | 67.2 | 46.3 | - | - | - | 4.4 | 15.5 | 52.4 | 15.7 | 39.2 | 61.0 | 17.9 | 46.7 | 65.9 |
| LLaVA-Next | 32.4 | 58.3 | 32.7 | - | - | - | 2.9 | 14.6 | 50.4 | 12.4 | 32.7 | 54.5 | 7.6 | 36.2 | 59.3 |
| DeepSeek R1 | 13.9 | 50.0 | 19.4 | - | - | - | 4.1 | 44.2 | 49.0 | 0.0 | 0.0 | 49.3 | 2.3 | 40.9 | 57.2 |
| Gemini 2.5 Pro | 16.0 | 51.1 | 21.4 | - | - | - | 2.4 | 20.0 | 53.8 | 5.1 | 25.0 | 48.1 | 4.2 | 32.7 | 45.2 |
| SleepLM (Cap) | **71.2** | 82.2 | 70.2 | - | - | - | 26.0 | 58.9 | 74.2 | 18.9 | 55.1 | **71.0** | 38.5 | 73.3 | 84.6 |
| SleepLM (CLIP) | 68.7 | 83.0 | 72.9 | - | - | - | - | - | - | - | - | - | - | - | - |
| SleepLM (CoCa) | 70.3 | **84.0** | **74.6** | - | - | - | 23.8 | 58.0 | 73.5 | **19.1** | 55.6 | 70.9 | **40.1** | 72.8 | 83.3 |
| SleepLM (ReCoCa) | 69.5 | 83.7 | 74.2 | - | - | - | **26.9** | **61.5** | **78.1** | 17.6 | 52.2 | 69.1 | 40.0 | **75.0** | **86.9** |
| *MROS:* | | | | | | | | | | | | | | | |
| Qwen3-VL-8B-Instruct | 48.0 | 68.6 | 49.1 | 4.8 | 9.5 | 52.6 | 7.2 | 22.4 | 54.7 | 29.1 | 61.7 | 62.7 | 19.0 | 51.1 | 67.7 |
| LLaVA-Next | 27.6 | 56.6 | 29.8 | 1.6 | 7.7 | 52.4 | 1.9 | 12.7 | 50.7 | 21.0 | 49.3 | 51.3 | 9.0 | 39.2 | 59.9 |
| DeepSeek R1 | 12.5 | 50.8 | 22.3 | 0.0 | 0.0 | 45.4 | 2.8 | 25.3 | 53.7 | 1.7 | 8.3 | 52.2 | 1.1 | 32.0 | 57.6 |
| Gemini 2.5 Pro | 17.0 | 52.1 | 21.9 | 1.2 | 10.5 | **82.7** | 0.0 | 11.8 | 52.4 | 7.7 | 39.5 | 52.4 | 3.2 | 31.0 | 55.7 |
| SleepLM (Cap) | 70.1 | 81.8 | 69.5 | 23.0 | **50.0** | 68.8 | 13.2 | 34.5 | 60.5 | 36.1 | 74.3 | 66.9 | 43.0 | 72.0 | 81.3 |
| SleepLM (CLIP) | 69.5 | 83.5 | 73.7 | - | - | - | - | - | - | - | - | - | - | - | - |
| SleepLM (CoCa) | 70.8 | 84.6 | 75.6 | 22.6 | 43.1 | 64.8 | 14.0 | 33.3 | 59.9 | 34.4 | 72.4 | 66.3 | 42.2 | 70.5 | 80.6 |
| SleepLM (ReCoCa) | **72.3** | **85.5** | **77.0** | **23.6** | 42.9 | 66.0 | **18.3** | **39.5** | **63.0** | **36.3** | **74.3** | **67.0** | **44.6** | **74.5** | **83.6** |
| *SHHS:* | | | | | | | | | | | | | | | |
| Qwen3-VL-8B-Instruct | 55.7 | 74.7 | 58.5 | 0.0 | 0.0 | 50.0 | 15.9 | 43.4 | 58.7 | 12.0 | 43.6 | 60.4 | 22.7 | 57.3 | 71.4 |
| LLaVA-Next | 39.3 | 61.9 | 38.5 | 0.0 | 0.0 | 49.5 | 6.0 | 30.3 | 52.5 | 10.3 | 36.0 | 52.6 | 8.5 | 40.7 | 61.3 |
| DeepSeek R1 | 14.9 | 51.9 | 20.7 | 0.0 | 0.0 | 42.0 | 4.6 | 57.6 | 58.6 | 0.7 | 8.0 | 51.5 | 0.9 | 26.8 | 42.5 |
| Gemini 2.5 Pro | 22.4 | 53.3 | 30.1 | 0.0 | 0.0 | 38.0 | 3.2 | 13.3 | 50.1 | 2.0 | 16.7 | 45.1 | 2.1 | 27.6 | 45.6 |
| SleepLM (Cap) | **76.8** | 85.8 | 76.4 | **25.2** | 48.3 | **72.3** | 35.7 | 66.3 | 74.1 | 18.5 | 51.6 | 66.3 | **43.7** | 75.7 | 84.6 |
| SleepLM (CLIP) | 72.3 | 85.6 | 77.2 | - | - | - | - | - | - | - | - | - | - | - | - |
| SleepLM (CoCa) | 73.0 | 85.9 | 77.8 | 13.6 | 37.5 | 64.3 | 33.0 | 66.0 | 74.0 | 17.0 | 50.7 | 65.6 | 42.1 | 74.7 | 84.4 |
| SleepLM (ReCoCa) | 74.6 | **87.1** | **79.6** | 24.7 | **51.0** | 70.8 | **37.6** | **69.2** | **76.4** | **22.4** | **57.8** | **70.3** | 42.8 | **76.8** | **85.7** |

As detailed in Table 15, ReCoCa demonstrates superior zero-shot capabilities, achieving the top performance across the majority of tasks and securing the second-best position in the remaining metrics. This consistent dominance highlights the efficacy of our proposed architecture over both generic multimodal baselines and larger proprietary models.

## D.6. Retrieval Raw Results

*Table 16.* **Zero-shot retrieval results across datasets.** † indicates retrieval evaluated on 100 samples due to context length constraints. Best is **bold**; second best is underlined.

| Model | Text→Signal | | | | | Signal→Text | | | | |
|---|---|---|---|---|---|---|---|---|---|---|
| | R@1 | R@5 | R@10 | Median Rank | Mean Rank | R@1 | R@5 | R@10 | Median Rank | Mean Rank |
| *CFS:* | | | | | | | | | | |
| DeepSeek R1[†] | - | - | - | - | - | 2.0 | 5.0 | 13.0 | - | - |
| Gemini 2.5 Pro[†] | - | - | - | - | - | 5.0 | 8.0 | 11.0 | - | - |
| SleepLM (CLIP) | 66.0 | 87.8 | 92.5 | 1.00 | 5.97 | 57.6 | 83.0 | 89.3 | 1.00 | 8.12 |
| SleepLM (CoCa) | 71.3 | 90.2 | 93.7 | 1.00 | **4.69** | 65.7 | **87.0** | **91.7** | 1.00 | **5.96** |
| SleepLM (ReCoCa) | **78.7** | **91.5** | **94.5** | 1.00 | 6.14 | **70.0** | 86.5 | 90.8 | 1.00 | 9.31 |
| *MROS:* | | | | | | | | | | |
| DeepSeek R1[†] | - | - | - | - | - | 2.0 | 6.0 | 13.0 | - | - |
| Gemini 2.5 Pro[†] | - | - | - | - | - | 6.0 | 17.0 | 28.0 | - | - |
| SleepLM (CLIP) | 67.8 | 83.5 | 87.4 | 1.00 | 8.39 | 68.7 | 85.4 | 90.3 | 1.00 | 5.05 |
| SleepLM (CoCa) | 70.5 | 83.3 | 87.7 | 1.00 | **7.54** | 71.4 | 86.1 | 91.0 | 1.00 | 4.36 |
| SleepLM (ReCoCa) | **72.7** | **84.5** | **88.3** | 1.00 | 7.91 | **76.2** | **88.2** | **92.5** | 1.00 | **4.03** |
| *SHHS:* | | | | | | | | | | |
| DeepSeek R1[†] | - | - | - | - | - | 1.0 | 6.0 | 11.0 | - | - |
| Gemini 2.5 Pro[†] | - | - | - | - | - | 1.0 | 7.0 | 13.0 | - | - |
| SleepLM (CLIP) | 89.1 | 98.9 | 99.6 | 1.00 | 1.28 | 90.7 | 99.1 | 99.6 | 1.00 | 1.23 |
| SleepLM (CoCa) | 90.9 | 99.0 | 99.7 | 1.00 | 1.23 | 92.9 | 99.5 | 99.7 | 1.00 | 1.19 |
| SleepLM (ReCoCa) | **96.1** | **99.8** | **100.0** | 1.00 | **1.07** | **96.7** | **99.8** | **100.0** | 1.00 | **1.06** |

In this section, we present the comprehensive, unaggregated results for cross-modal retrieval. Note that due to context window limitations, proprietary LLMs are evaluated on a reduced pool size of $N = 100$, whereas all SleepLM variants are tested on a significantly more challenging pool of $N = 2000$. Despite this structural advantage, the LLM baselines yield performance near random chance, falling far behind the specialized pretraining variants. As detailed in Table 16, ReCoCa consistently achieves top-tier performance across the majority of metrics, validating its superior semantic alignment even when compared against baselines operating under easier test conditions.

## D.7. Regression Raw Results

In this section, we present the comprehensive, unaggregated results for zero-shot regression, categorized into two distinct tasks:

Explicit Signal Grounding (Channel Stats): To assess the model's understanding of visible input signals, we parse statistical descriptors from the generated channel-specific captions. We calculate the MAE and sMAPE for each statistic individually.

Implicit Physiological Inference (HR & SpO2): We isolate Heart Rate (HR) and Blood Oxygen (SpO2) results as these signals are explicitly excluded from the encoder input.

- **Scalar Estimation:** We evaluate the model's ability to estimate (Max, Min, Mean) for HR and (Mean, Min) for SpO2 based solely on cross-channel correlations (e.g., deriving heart rate from ECG).

- **Trend Detection:** We calculate the Recall for estimated morphological changes (trends) in these implicit signals.

Analysis: As shown in the results, proprietary LLMs demonstrate remarkably strong performance on HR and SpO2 estimation. This suggests that these reasoning-heavy models effectively leverage well-established clinical algorithms (e.g., QRS detection logic) to derive vital signs from related inputs like ECG. However, ReCoCa maintains highly competitive

*Table 17.* **Zero-shot regression and trend statistics results across datasets.** Best is **bold**; second best is underlined.

| Model | Heart Rate | | | | SpO$_2$ | | | Channel Stats |
|---|---|---|---|---|---|---|---|---|
| | Mean MAE$^{\downarrow}$ | Max MAE$^{\downarrow}$ | Min MAE$^{\downarrow}$ | Trend Recall$^{\uparrow}$ | Mean MAE$^{\downarrow}$ | Min MAE$^{\downarrow}$ | Trend Recall$^{\uparrow}$ | Avg sMAPE$^{\downarrow}$ |
| *CFS:* | | | | | | | | |
| Qwen-VL-8B-Instruct | 18.07 | 23.56 | 14.69 | 11.8 | 2.13 | 3.46 | 10.0 | 33.56 |
| LLaVA-Next | 17.98 | 23.21 | 16.77 | 18.3 | 2.56 | 4.73 | 16.7 | 37.41 |
| DeepSeek R1 | 8.54 | 18.06 | 11.08 | **28.6** | **1.84** | 3.57 | 9.6 | - |
| Gemini 2.5 Pro | **2.70** | **4.96** | **6.19** | 14.2 | 2.44 | **2.73** | 12.7 | - |
| SleepLM (Cap) | 3.57 | 6.23 | 9.79 | 18.4 | 2.40 | 3.82 | 35.4 | 8.85 |
| SleepLM (CoCa) | 3.25 | 5.67 | 9.74 | 19.7 | 2.42 | 3.89 | **42.8** | 9.02 |
| SleepLM (ReCoCa) | 3.11 | 5.41 | 9.97 | 20.0 | 2.52 | 4.15 | 40.8 | **3.99** |
| *SHHS:* | | | | | | | | |
| Qwen-VL-8B-Instruct | 9.44 | 10.07 | 9.69 | 32.6 | 2.15 | 3.16 | 26.9 | 28.63 |
| LLaVA-Next | 11.37 | 12.20 | 12.16 | 26.0 | 2.60 | 4.44 | 20.9 | 35.81 |
| DeepSeek R1 | 10.30 | 20.19 | 11.35 | 26.0 | **1.93** | 5.91 | 9.3 | - |
| Gemini 2.5 Pro | 1.32 | 8.12 | 6.06 | 7.2 | 1.96 | **1.78** | 11.5 | - |
| SleepLM (Cap) | 0.86 | 1.22 | 1.72 | 51.1 | 2.00 | 3.10 | 37.2 | 6.37 |
| SleepLM (CoCa) | 0.85 | 1.19 | **1.63** | 50.2 | 1.99 | 3.02 | **41.6** | 6.51 |
| SleepLM (ReCoCa) | **0.84** | **1.18** | 1.64 | **51.6** | 1.95 | 2.94 | 37.5 | **2.73** |

performance on these implicit tasks while achieving a dominant margin on the "Channel Stats" metric, which reports the averaged sMAPE across all these signal descriptor excpet HR and SpO2 trend. This confirms that while LLMs excel at specific, algorithmic derivations, `ReCoCa` possesses a superior, generalized grounding in the raw signal morphology across the full montage.

## D.8. Statistical Results for Full Night Metrics

*Table 18.* **Full-Night PSG Metrics Comparison: CoCa vs Qwen3 (5 SHHS Subjects).** $\Delta = |\text{Qwen}-\text{Doc}| - |\text{CoCa}-\text{Doc}|$. Positive values (bold) indicate CoCa is closer to ground truth.

| Metric | Doc GT | CoCa | Qwen3 | CoCa−Doc | Qwen−Doc | $\Delta$ |
|---|---|---|---|---|---|---|
| Sleep Efficiency (%) | 87.95 | 88.38 | 89.28 | +0.42 | +1.33 | **+0.90** |
| Sleep Latency (min) | 17.40 | 14.50 | 4.30 | -2.90 | -13.10 | **+10.20** |
| WASO (min) | 38.40 | 39.40 | 44.30 | +1.00 | +5.90 | **+4.90** |
| Arousal Index (all) | 23.04 | 28.01 | 20.64 | +4.97 | -2.41 | -2.56 |
| Arousal Index (NREM) | 24.12 | 26.10 | 19.29 | +1.98 | -4.83 | **+2.84** |
| Central Apnea Index | 0.21 | 0.11 | 0.13 | -0.10 | -0.08 | -0.02 |
| CAI (4% desat) | 0.08 | 0.00 | 0.00 | -0.08 | -0.08 | +0.00 |
| CAI (4% desat + arousal) | 0.08 | 0.04 | 0.10 | -0.03 | +0.03 | -0.00 |
| AHI (3% desat) | 25.02 | 16.01 | 6.69 | -9.01 | -18.33 | **+9.32** |
| AHI (3% desat + arousal) | 26.97 | 23.52 | 17.02 | -3.45 | -9.95 | **+6.50** |
| AHI (4% desat) | 21.91 | 11.21 | 3.55 | -10.70 | -18.36 | **+7.66** |
| AHI (4% desat + arousal) | 24.31 | 20.99 | 15.80 | -3.33 | -8.51 | **+5.19** |
| RDI (no desat) | 37.46 | 38.02 | 28.71 | +0.56 | -8.75 | **+8.19** |
| RDI (arousal only) | 12.93 | 18.94 | 15.05 | +6.01 | +2.12 | -3.89 |

In this section, we present the quantitative evaluation of full-night clinical variable estimation. To assess real-world utility, we randomly sample 5 full-night recordings from the SHHS dataset. We process each recording using a sliding window approach to generate fine-grained, epoch-level predictions for sleep stages, event locations, and vital signal statistics. These temporal outputs are then aggregated to compute a comprehensive suite of clinically relevant indices.

Table 18 provides the complete comparison between `ReCoCa` and the fine-tuned Qwen3-VL-8B-Instruct. We observe the following key trends:

- Sleep Architecture: `ReCoCa` dominates on macro-structural metrics, including Sleep Efficiency (`slpeffp`), Sleep Latency (`slplatp`), and Wake After Sleep Onset (`waso`).

- Respiratory Indices: `ReCoCa` achieves significantly lower error rates on Apnea-Hypopnea Index (AHI) metrics (e.g., `ahi_a0h3`, `ahi_a0h4`) and their arousal-linked variants. Notably, it achieves nearly perfect performance on the Respiratory Disturbance Index (`RDI0P`), showing a delta of +0.56 compared to Qwen3's deviation of -8.75.

- Arousal Metrics: Qwen3 demonstrates marginally better performance on the aggregate Arousal Index (`ai_all`) and specific Central Apnea Index (CAI) sub-metrics, though it fails to maintain consistency across the broader respiratory reporting.

This evaluation confirms that `ReCoCa` successfully translates epoch-level understanding into accurate, longitudinal clinical summaries, outperforming larger fine-tuned VLMs in generating consistent full-night reports.

*Table 19.* Example templated sleep report generated from our predicted variables.

**Example templated sleep report (Patient: shhs1-201545).**
**Full-night summary:** Wake time accounts for 12.3%, sleep time accounts for 87.7%. AHI=60.91 events/hr, $ODI_4$=46.94 events/hr. Mean $SpO_2$=93.7%, minimum $SpO_2$=40.0%. Event composition: hypopnea 100.0%, central apnea 0.0%.

| **Sleep structure** | |
| --- | --- |
| Total recording time (TRT) | 08:09:30 |
| Total sleep time (TST) | 07:09:30 |
| Sleep efficiency (SE) | 87.7% |
| Wake after sleep onset (WASO) | 59.5 min |

| **Respiratory events (from generated variables)** | **Count** | **Index (events/hr)** |
| --- | --- | --- |
| Central apnea | 0 | 0.00 |
| Hypopnea | 436 | 60.91 |
| **Total (all resp. events)** | 436 | 60.91 |
| Oxygen desaturation ($\geq$4%) | 336 | 46.94 |

| **Oxygen saturation ($SpO_2$)** | |
| --- | --- |
| Mean $SpO_2$ (TRT) | 93.7% |
| Minimum $SpO_2$ (TRT) | 40.0% |

### D.9. Full Generation Quality Results

In this section, we extend our qualitative assessment to the finetuned vision-language baselines, Qwen3-VL-8B-Instruct and LLaVA-Next. As illustrated in Fig. 9, while these specialized models generate marginally more accurate descriptions than the LLMs, they fail to close the performance gap. Both models remain prone to significant misconceptions and lack the granular spatial awareness required for precise event localization.

### D.10. Targeted Generation Ability

See Figure 10 as a demonstration for `SleepLM`'s targeted generation ability.

### D.11. Caption Generation performance

We provide the raw generation performance metrics in this section to compare the caption quality generated by `SleepLM` and other LLM and finetuned VLM baselines. See Table 20 for the raw scores. `SleepLM` outperforms the baselines across

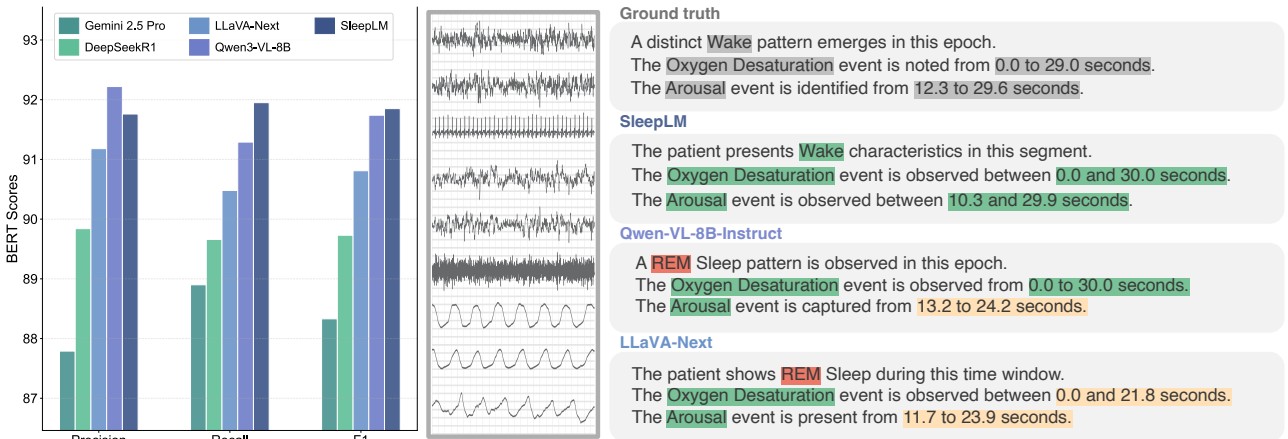

*Figure 9.* Full generation from Qwen3 and DeepSeek-R1 from the paper example.

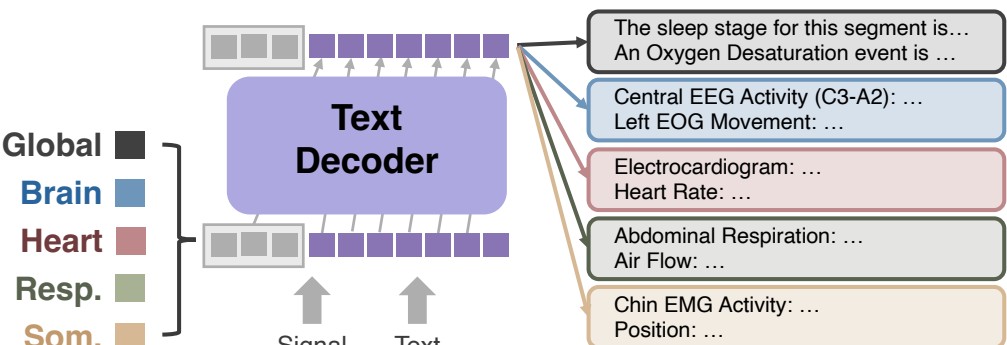

*Figure 10.* **Illustration of targeted generation:** Given a pair of input signal and text, our model is able to perform precise, targeted generations on one or multiple designated modalities by prepending corresponding condition tokens during text decoding.

all evaluated metrics, demonstrating a more accurate caption quality.

*Table 20.* **Caption generation performance comparison between** `SleepLM` **and baselines.**

| Model | BERTPrecision | BERTRecall | BERTF1 | METEOR |
|---|---|---|---|---|
| DeepSeekR1 | 89.8 | 89.7 | 89.7 | 32.5 |
| Gemini2.5pro | 87.8 | 88.9 | 88.3 | 33.0 |
| LLaVA-Next | 91.2 | 90.5 | 90.8 | 42.3 |
| `SleepLM` | **91.8** | **92.0** | **91.8** | **44.2** |

# E. Additional Examples

We provide extra qualitative examples in this section for reference purposes.

## E.1. Multilevel Caption Examples

**Examples Captions Generated by Our Data Pipeline (Categorized by Modalities)**

```
==================================== Example 1 ====================================

SLEEP STAGE: During this epoch, the patient exhibits Stable Light Sleep (N2).

SLEEP EVENT: The Hypopnea event manifests from 0.0 to 3.9 seconds. The Oxygen Desaturation event is
    observed between 0.0 and 30.0 seconds. Analysis reveals a Arousal event from 3.5 to 8.5 seconds.
    A Hypopnea event is noted between 24.3 and 30.0 seconds.

BRAIN: Left EOG Movement: slow eye movement relative power measuring roughly 30%, REM saccadic
    relative power measuring 53.16%. Right EOG Movement: slow eye movement relative power equal to
    28.50%, REM saccadic relative power valued at 55.23%. Central EEG Activity (C3-A2): delta
    relative power valued at 75.83%, combined with theta relative power of 9.82%, along with alpha
    relative power equal to approximately 8%, beta relative power at 4.07%. Central EEG Activity
    (C4-A1): delta relative power at 69.12%, plus theta relative power valued at 11.11%, a alpha
    relative power of 13.49%, and a beta relative power of roughly 4%.

HEART: Electrocardiogram: estimated beat rate valued at 78.2 bpm, ultra-short HRV (RMSSD) valued at
    21.41 ms. Heart Rate: minimum value equal to 75.20, together with a maximum value at 85.20, mean
    valued at 78.51, an decreasing tendency from 0.9 to 5.6 seconds, along with a increasing pattern
    from 6.6 to 15.0 seconds, along with an decreasing behavior from 15.9 to 24.4 seconds. Blood
    Oxygen Saturation: a minimum value valued at 89.31, combined with a mean equal to 91.87,
    combined with a decreasing interval from 0.9 to 18.8 seconds, together with a increasing
    progression from 19.7 to 30.0 seconds.

RESPIRATORY: Abdominal Respiration: respiratory rate measuring 18.73 bpm, and lastly a breath
    interval variability of 0.70 s. Thoracic Respiration: respiratory rate at 18.11 bpm, and breath
    interval variability equal to 0.80 s. Air Flow: airflow rate measuring 18.55 bpm, a inspiratory
    flow flatness equal to 0.65.

SOMATIC: Chin EMG Activity: a median frequency measuring 16.69 Hz, along with burst intensity ratio
    of 2.11, and finally burst modulation rate of 0.98 Hz. The patient stays lying supine for the
    full epoch.

==================================== Example 2 ====================================

SLEEP STAGE: A distinct Wake pattern emerges in this epoch.

SLEEP EVENT: The epoch showed no notable sleep disruptions.
```

BRAIN: Left EOG Movement: slow eye movement relative power of 25.87%, plus REM saccadic relative power equal to 39.43%. Right EOG Movement: slow eye movement relative power of 27.01%, REM saccadic relative power of 40.66%. Central EEG Activity (C3-A2): a delta relative power of 20.64%, as well as a theta relative power valued at around 15%, along with a alpha relative power measuring roughly 32%, and beta relative power valued at around 29%. Central EEG Activity (C4-A1): delta relative power valued at 24.38%, plus a theta relative power at approximately 12%, including alpha relative power valued at 31.34%, beta relative power measuring 28.54%.

HEART: Electrocardiogram: estimated beat rate valued at 83.0 bpm, a ultra-short HRV (RMSSD) valued at 10.91 ms. Heart Rate: a minimum value equal to 82.23, plus a maximum value measuring 85.36, together with a mean equal to 83.73, including a increasing interval from 0.9 to 3.8 seconds, a decreasing period from 4.7 to 11.3 seconds, together with a increasing movement from 12.2 to 15.0 seconds, together with an increasing segment from 15.9 to 20.6 seconds, plus a decreasing movement from 19.7 to 26.3 seconds, combined with an increasing trend from 27.2 to 30.0 seconds. Blood Oxygen Saturation: minimum value valued at 96.09, accompanied by mean equal to 97.16, in addition to a decreasing period from 0.9 to 15.0 seconds, and also an increasing behavior from 10.3 to 30.0 seconds.

RESPIRATORY: Abdominal Respiration: a respiratory rate at 12.21 bpm, plus breath interval variability of 1.83 s. Thoracic Respiration: respiratory rate valued at 11.93 bpm, and a breath interval variability equal to 1.84 s. Air Flow: airflow rate measuring 11.36 bpm, inspiratory flow flatness equal to 0.55.

SOMATIC: Chin EMG Activity: median frequency measuring 17.87 Hz, burst intensity ratio of 1.68, along with burst modulation rate measuring 1.03 Hz. The patient lies prone throughout the recording.

======================================= Example 3 =======================================

SLEEP STAGE: A clear REM Sleep pattern is detected in this epoch.

SLEEP EVENT: A Arousal event is present between 0.0 and 9.2 seconds. We detect a Hypopnea event from 16.3 to 30.0 seconds.

BRAIN: Left EOG Movement: slow eye movement relative power valued at 25.54%, and a REM saccadic relative power of 57.11%. Right EOG Movement: a slow eye movement relative power measuring 31.46%, and lastly a REM saccadic relative power measuring around 47%. Central EEG Activity (C3-A2): a delta relative power valued at 41.63%, accompanied by theta relative power of 10.02%, including a alpha relative power of 11.44%, and beta relative power valued at 34.41%. Central EEG Activity (C4-A1): delta relative power measuring approximately 54%, including theta relative power equal to 12.56%, alpha relative power of 7.81%, and beta relative power of 22.95%.

HEART: Electrocardiogram: a estimated beat rate of 48.9 bpm, as well as ultra-short HRV (RMSSD) measuring 85.54 ms. Heart Rate: minimum value measuring roughly 44, as well as a maximum value at 66.21, including mean valued at 53.96, a increasing pattern from 0.9 to 7.5 seconds, and finally an decreasing tendency from 10.3 to 20.6 seconds. Blood Oxygen Saturation: a minimum value equal to 92.19, in addition to a mean of roughly 94, as well as a decreasing period from 0.9 to 9.4 seconds, and lastly a increasing pattern from 4.7 to 28.1 seconds.

RESPIRATORY: Abdominal Respiration: respiratory rate measuring 15.90 bpm, and also a breath interval variability of 0.20 s. Thoracic Respiration: respiratory rate valued at 15.61 bpm, and breath interval variability valued at 0.41 s. Air Flow: a airflow rate equal to 17.41 bpm, in addition to a inspiratory flow flatness measuring 0.70.

SOMATIC: Chin EMG Activity: a median frequency valued at 15.05 Hz, burst intensity ratio at 3.05, a burst modulation rate valued at 0.78 Hz. The patient maintains the right position for this period.

## E.2. Prompts Used for LLM

Zeroshot Evaluation Prompt

```
You are an expert sleep medicine AI assistant analyzing polysomnography (PSG) signals.

## INPUT FORMAT
You will receive raw biosignal data from a 30-second sleep epoch. The signals are sampled at 64 Hz
  (1920 samples per channel).

The channels provided are:
  - ECG: Electrocardiogram - heart electrical activity
  - ABD: Abdominal respiration belt - breathing movement
  - THX: Thoracic respiration belt - chest breathing movement
  - AF: Airflow - nasal/oral airflow signal
  - EOG_E1_A2: Left electrooculogram - left eye movement
  - EOG_E2_A1: Right electrooculogram - right eye movement
  - EEG_C3_A2: Central EEG (C3-A2) - brain electrical activity
  - EEG_C4_A1: Central EEG (C4-A1) - brain electrical activity
  - EMG_Chin: Chin electromyogram - muscle activity
  - POS: Body position - somatic state

The data is formatted as a CSV table with columns for time and each channel:
Time(s),ECG,ABD,THX,AF,...
0.0000,-0.49,-0.76,0.12,...
0.0156,-0.55,-0.78,0.15,...
Each row represents a single time point with all channel values.

## YOUR TASK
Analyze the signals and provide:

1. **Sleep Stage**: Classify the epoch into one of these stages:
   - Wake
   - Light Sleep (N1)
   - Stable Light Sleep (N2)
   - Deep Sleep (N3)
   - REM Sleep

2. **Sleep Events**: Detect any of these events with their start and end times (in seconds from 0.0
   to 30.0):
   - Central Apnea
   - Hypopnea
   - Oxygen Desaturation
   - Arousal

   Look for patterns like:
   - Reduced airflow (AF channel) for apnea/hypopnea
   - Oxygen desaturation patterns (infer from overall signal changes)
   - Arousal patterns (sudden changes in EEG/EMG)

3. **Heart Rate Trends**: Analyze the ECG channel to detect periods of:
   - "increasing" heart rate
   - "decreasing" heart rate
   - "stable" heart rate
   Provide start and end times for each trend segment.

4. **SpO2 (Blood Oxygen) Trends**: Based on the overall physiological patterns, infer:
   - "increasing" blood oxygen
   - "decreasing" blood oxygen
   - "stable" blood oxygen
   Provide start and end times for each trend segment.

## IMPORTANT NOTES
- All times must be between 0.0 and 30.0 seconds
```

```
- If you don't detect any events, return an empty list
- If you cannot determine trends, return an empty list
- Provide confidence scores (0.0 to 1.0) for all predictions
- For trends, focus on detecting the main patterns - you don't need to capture every tiny
  fluctuation

## OUTPUT FORMAT
Respond with a JSON object matching the specified schema.
```

## Zeroshot Retrieval Prompt

```
You are an expert sleep medicine AI assistant performing signal-to-text retrieval.

## INPUT FORMAT
You will receive:
1. Raw polysomnography (PSG) biosignal data from a 30-second sleep epoch
2. A list of candidate text captions describing different sleep epochs

The signals are sampled at 64 Hz (1920 samples per channel).

The channels provided are:
  - ECG: Electrocardiogram - heart electrical activity
  - ABD: Abdominal respiration belt - breathing movement
  - THX: Thoracic respiration belt - chest breathing movement
  - AF: Airflow - nasal/oral airflow signal
  - EOG_E1_A2: Left electrooculogram - left eye movement
  - EOG_E2_A1: Right electrooculogram - right eye movement
  - EEG_C3_A2: Central EEG (C3-A2) - brain electrical activity
  - EEG_C4_A1: Central EEG (C4-A1) - brain electrical activity
  - EMG_Chin: Chin electromyogram - muscle activity
  - POS: Body position - somatic state

The signal data is formatted as a CSV table with columns for time and each channel:
Time(s),ECG,ABD,THX,AF,...
0.0000,-0.49,-0.76,0.12,...
0.0156,-0.55,-0.78,0.15,...

## YOUR TASK
Analyze the biosignal data and rank the captions by how well they describe this specific signal
  epoch.

Consider:
- Sleep stage characteristics (EEG patterns, eye movements, muscle tone)
- Respiratory events (airflow, chest/abdominal movement patterns)
- Cardiac patterns (heart rate variability)
- Temporal patterns and event timing
- Overall physiological coherence

## OUTPUT FORMAT
Respond with a JSON object containing:
- top_captions: List of up to 10 best matching captions, ranked from best to worst
  - Each entry has: caption_idx (0-based index) and confidence (0.0 to 1.0)
  - MUST include at least the top caption, preferably top 10
- reasoning: Brief explanation of your ranking

Be precise and analytical in your ranking.
```

## Unseen classification Prompt

```
You are an expert sleep medicine AI assistant performing signal-to-text retrieval.

## INPUT FORMAT
```

```
You will receive:
1. Raw polysomnography (PSG) biosignal data from a 30-second sleep epoch
2. A list of candidate text captions describing different sleep epochs

The signals are sampled at 64 Hz (1920 samples per channel).

The channels provided are:
  - ECG: Electrocardiogram - heart electrical activity
  - ABD: Abdominal respiration belt - breathing movement
  - THX: Thoracic respiration belt - chest breathing movement
  - AF: Airflow - nasal/oral airflow signal
  - EOG_E1_A2: Left electrooculogram - left eye movement
  - EOG_E2_A1: Right electrooculogram - right eye movement
  - EEG_C3_A2: Central EEG (C3-A2) - brain electrical activity
  - EEG_C4_A1: Central EEG (C4-A1) - brain electrical activity
  - EMG_Chin: Chin electromyogram - muscle activity
  - POS: Body position - somatic state

The signal data is formatted as a CSV table with columns for time and each channel:
Time(s),ECG,ABD,THX,AF,...
0.0000,-0.49,-0.76,0.12,...
0.0156,-0.55,-0.78,0.15,...

## YOUR TASK
Determine if **Obstructive Apnea** is present (1) or absent (0) in this epoch.

## OUTPUT FORMAT
Respond with a JSON object containing:
- prediction: 0 or 1
- confidence: your confidence score (0.0 to 1.0)
- reasoning: brief explanation of your decision
```

## E.3. Templates Examples for Classification

```
Templates Pool

1. Templates for embedding similarity based sleep stage classification

SLEEP_STAGE_TEMPLATES = [
    "The sleep stage for this epoch is {sleep_stage}.",
    "This epoch is classified as {sleep_stage}.",
    "The patient is in {sleep_stage} during this period.",
    "Sleep stage analysis shows {sleep_stage} for this epoch.",
    "This 30-second segment corresponds to {sleep_stage}.",
    "The sleep stage classification indicates {sleep_stage}.",
    "During this epoch, the patient exhibits {sleep_stage}.",
    "Sleep stage assessment reveals {sleep_stage} for this period.",
    "This time window is characterized by {sleep_stage}.",
    "The sleep stage for this segment is {sleep_stage}.",
    "The patient demonstrates {sleep_stage} during this epoch.",
    "A {sleep_stage} pattern is observed in this epoch.",
    "The sleep stage analysis reveals {sleep_stage} for this period.",
    "This epoch shows evidence of {sleep_stage}.",
    "The patient presents {sleep_stage} characteristics in this segment.",
    "A clear {sleep_stage} pattern is detected in this epoch.",
    "The patient manifests {sleep_stage} during this time window.",
    "Sleep stage evaluation indicates {sleep_stage} for this epoch.",
    "The patient displays {sleep_stage} behavior in this segment.",
    "A distinct {sleep_stage} pattern emerges in this epoch.",
    "The patient exhibits {sleep_stage} during this period.",
    "Sleep stage assessment shows {sleep_stage} for this epoch.",
```

```
    "The patient reveals {sleep_stage} characteristics in this segment.",
    "A {sleep_stage} state is observed during this epoch.",
    "The patient demonstrates {sleep_stage} patterns in this period.",
    "Sleep stage analysis indicates {sleep_stage} for this epoch.",
    "The patient shows {sleep_stage} during this time window.",
    "A {sleep_stage} phase is detected in this epoch.",
    "The patient exhibits {sleep_stage} features in this segment.",
    "Sleep stage evaluation reveals {sleep_stage} for this epoch.",
    "The patient presents {sleep_stage} during this period.",
    "A {sleep_stage} condition is observed in this epoch.",
]

2. Templates for embedding similarity based unseen events classification:

PRESENCE_TEMPLATES = [
    "a {event_name} event is present during this epoch.",
    "the {event_name} event occurs in this period.",
    "we observe a {event_name} event.",
    "a {event_name} event is detected.",
    "the {event_name} event is identified in this epoch.",
    "this epoch contains a {event_name} event.",
    "a {event_name} event is recorded during this period.",
    "the presence of a {event_name} event is noted.",
    "we detect a {event_name} event in this epoch.",
    "a {event_name} event is observed.",
    "the {event_name} event manifests during this period.",
    "analysis reveals a {event_name} event.",
    "a {event_name} event occurs in this recording.",
    "the {event_name} event is present.",
    "we identify a {event_name} event.",
    "this period shows a {event_name} event.",
    "a {event_name} event is documented.",
    "the {event_name} event is captured in this epoch.",
    "we observe the occurrence of a {event_name} event.",
    "a {event_name} event is evident during this period.",
    "the {event_name} event appears in this epoch.",
    "this epoch exhibits a {event_name} event.",
    "a {event_name} event is noted.",
    "the {event_name} event is found in this recording.",
    "we record a {event_name} event during this period.",
    "a {event_name} event is visible in this epoch.",
    "the {event_name} event is apparent.",
    "this period includes a {event_name} event.",
    "a {event_name} event is present in the recording.",
    "the {event_name} event is confirmed in this epoch.",
]

ABSENCE_TEMPLATES = [
    "no significant sleep events were detected during this epoch.",
    "the epoch showed no notable sleep disruptions.",
    "sleep remained undisturbed throughout the epoch.",
    "the patient experienced uninterrupted sleep during this period.",
    "sleep continuity was maintained without events.",
    "no respiratory or movement-related sleep disturbances were observed.",
    "the recording showed no significant sleep-related events.",
    "no pathological sleep events were identified in this epoch.",
    "sleep proceeded without notable interruptions or events.",
    "the patient maintained stable sleep without events.",
    "no clinically significant sleep events were recorded.",
    "sleep architecture remained undisturbed during this period.",
    "the epoch was characterized by uninterrupted sleep patterns.",
    "no sleep-disordered breathing events were detected.",
```

```
    "this epoch is free of significant sleep events.",
    "the recording shows normal sleep without events.",
    "no abnormal sleep events were found in this period.",
    "sleep was uninterrupted by events.",
    "the epoch demonstrates absence of sleep events.",
    "no respiratory events or arousals were detected.",
    "the patient slept without disturbances.",
    "this period lacks any significant sleep events.",
    "no events were observed during this epoch.",
    "sleep quality was maintained without events.",
    "the recording is negative for sleep events.",
    "no clinically relevant events were identified.",
    "this epoch shows normal sleep patterns without events.",
    "the analysis found no pathological events.",
    "the absence of sleep events characterizes this period.",
    "no pathological findings were detected.",
]
```

## E.4. More Retrieval Quality Examples

### More retrieval Examples

```
===================================== Paper Example =====================================

Original Caption: "A REM Sleep pattern is observed in this epoch. The Oxygen Desaturation event is
  documented from 13.0 to 28.0 seconds. The Hypopnea event is observed from 20.0 to 30.0 seconds."

MOST SIMILAR:
    1. "A REM Sleep pattern is observed in this epoch. The Oxygen Desaturation event is documented
       from 13.0 to 28.0 seconds. The Hypopnea event is observed from 20.0 to 30.0 seconds."
    2. "The sleep stage for this segment is REM Sleep. The Hypopnea event is recorded from 0.0 to
       19.2 seconds. A Oxygen Desaturation event occurs from 17.6 to 30.0 seconds. A Hypopnea event
       is identified between 25.8 and 30.0 seconds."
    3. "During this epoch, the patient exhibits REM Sleep. We observe a Hypopnea event between 0.0
       and 3.9 seconds. The Hypopnea event spans 16.1 to 28.3 seconds. The Oxygen Desaturation
       event is observed from 17.3 to 30.0 seconds."

INTERMMEDIATE SIMILAR:
    1. "Sleep stage evaluation reveals REM Sleep for this epoch. A Oxygen Desaturation event is
       observed between 15.1 and 27.0 seconds. The Hypopnea event is documented from 20.9 to 30.0
       seconds."
    2. "The patient exhibits Light Sleep (N1) features in this segment. The patient maintained
       stable sleep without events."
    3. "The sleep stage classification indicates Deep Sleep (N3). No significant sleep events were
       detected during this epoch."

LEAST SIMILAR:
    1. "The patient shows Deep Sleep (N3) during this time window. No sleep-disordered breathing
       events were detected."
    2. "The patient is in Stable Light Sleep (N2) during this period. No clinically significant
       sleep events were recorded."
    3. "Sleep stage evaluation reveals Stable Light Sleep (N2) for this epoch. No sleep-disordered
       breathing events were detected."

===================================== Extra Example 1 =====================================

Original Caption: "The patient exhibits Stable Light Sleep (N2) features in this segment. No
  sleep-disordered breathing events were detected."

MOST SIMILAR:
```

    1. "The patient exhibits Stable Light Sleep (N2) features in this segment. No sleep-disordered breathing events were detected."
    2. "This time window is characterized by Stable Light Sleep (N2). No respiratory or movement-related sleep disturbances were observed."
    3. "Sleep stage analysis shows Stable Light Sleep (N2) for this epoch. The epoch showed no notable sleep disruptions."

INTERMMEDIATE SIMILAR:
    1. "The patient displays Stable Light Sleep (N2) behavior in this segment. Sleep proceeded without notable interruptions or events."
    2. "The patient displays Stable Light Sleep (N2) behavior in this segment. No significant sleep events were detected during this epoch."
    3. "The patient exhibits Stable Light Sleep (N2) during this period. A Hypopnea event is detected between 9.1 and 29.3 seconds."

LEAST SIMILAR:
    1. "The patient presents REM Sleep during this period. The Hypopnea event is present between 29.3 and 30.0 seconds."
    2. "The patient presents Stable Light Sleep (N2) characteristics in this segment. The Oxygen Desaturation event is documented from 0.0 to 12.0 seconds. A Oxygen Desaturation event is noted between 0.0 and 16.0 seconds. The Hypopnea event is observed between 18.0 and 30.0 seconds. A Oxygen Desaturation event occurs from 21.0 to 30.0 seconds."
    3. "The patient shows REM Sleep during this time window. The Oxygen Desaturation event is documented from 0.0 to 30.0 seconds. We observe a Oxygen Desaturation event between 14.0 and 30.0 seconds."

================================== Extra Example 2 ==================================

Original Caption: "This 30-second segment corresponds to Stable Light Sleep (N2). Sleep continuity was maintained without events."

MOST SIMILAR:
    1. "This 30-second segment corresponds to Stable Light Sleep (N2). Sleep continuity was maintained without events."
    2. "During this epoch, the patient exhibits Stable Light Sleep (N2). No sleep-disordered breathing events were detected."
    3. "Sleep stage analysis indicates Stable Light Sleep (N2) for this epoch. The recording showed no significant sleep-related events."

INTERMEDIATE SIMILAR:
    1. "The sleep stage classification indicates Stable Light Sleep (N2). The recording showed no significant sleep-related events."
    2. "The sleep stage analysis reveals Stable Light Sleep (N2) for this period. A Hypopnea event is detected from 0.0 to 2.8 seconds. Analysis reveals a Oxygen Desaturation event from 4.7 to 30.0 seconds."
    3. "The sleep stage for this segment is Stable Light Sleep (N2). The Arousal event is documented from 17.9 to 26.1 seconds."

LEAST SIMILAR:
    1. "The patient displays Stable Light Sleep (N2) behavior in this segment. A Hypopnea event is noted between 0.0 and 13.0 seconds. The Arousal event is identified from 11.6 to 17.3 seconds. A Oxygen Desaturation event is identified between 20.0 and 30.0 seconds."
    2. "The sleep stage for this segment is REM Sleep. The Hypopnea event is recorded from 0.0 to 19.2 seconds. A Oxygen Desaturation event occurs from 17.6 to 30.0 seconds. A Hypopnea event is identified between 25.8 and 30.0 seconds."
    3. "Sleep stage evaluation indicates REM Sleep for this epoch. The Oxygen Desaturation event manifests from 0.0 to 17.1 seconds. The Oxygen Desaturation event is noted from 14.0 to 30.0 seconds."

================================== Extra Example 3 ==================================

```
Original Caption: "Sleep stage assessment reveals Stable Light Sleep (N2) for this period. The
  Hypopnea event is recorded from 0.0 to 10.7 seconds. The Hypopnea event is present from 21.2 to
  30.0 seconds."

MOST SIMILAR:
    1. "Sleep stage assessment reveals Stable Light Sleep (N2) for this period. The Hypopnea event
       is recorded from 0.0 to 10.7 seconds. The Hypopnea event is present from 21.2 to 30.0
       seconds."
    2. "The patient displays Stable Light Sleep (N2) behavior in this segment. The Hypopnea event
       is documented from 4.0 to 14.0 seconds."
    3. "Sleep stage assessment shows Stable Light Sleep (N2) for this epoch. We observe a Hypopnea
       event between 26.5 and 30.0 seconds."

INTERMEDIATE SIMILAR:
    1. "The patient shows REM Sleep during this time window. The Hypopnea event lasts from 9.0 to
       26.0 seconds. The Arousal event is captured from 26.9 to 30.0 seconds."
    2. "The sleep stage for this segment is REM Sleep. The patient experienced uninterrupted sleep
       during this period."
    3. "A Stable Light Sleep (N2) state is observed during this epoch. The recording showed no
       significant sleep-related events."

LEAST SIMILAR:
    1. "The sleep stage for this segment is Wake. No clinically significant sleep events were
       recorded."
    2. "A distinct Stable Light Sleep (N2) pattern emerges in this epoch. Sleep architecture
       remained undisturbed during this period."
    3. "Sleep stage evaluation indicates Deep Sleep (N3) for this epoch. Sleep proceeded without
       notable interruptions or events."
```

## E.5. More Perturbation Study

**More Perturbation Examples**

```
================================= Paper Example =================================

Similarity = 0.3114 (original)
"A REM Sleep phase is detected in this epoch. The Hypopnea event is documented from 0.0 to 5.6
  seconds. The Oxygen Desaturation event lasts from 14.2 to 30.0 seconds. The Hypopnea event is
  present from 15.4 to 30.0 seconds."

Similarity = 0.2743
"... The Hypopnea event is documented from 0.0 to 2.0 seconds. ... The Hypopnea event is documented
  from 28.0 to 30.0 seconds."

Similarity = 0.3195
"... The Hypopnea event is documented from 0.0 to 4.0 seconds. ... The Hypopnea event is documented
  from 22.0 to 30.0 seconds."

Similarity = 0.3511
"... The Hypopnea event is documented from 0.0 to 6.0 seconds. ... The Hypopnea event is documented
  from 16.0 to 30.0 seconds."

Similarity = 0.3152
"... The Hypopnea event is documented from 0.0 to 8.0 seconds. ... The Hypopnea event is documented
  from 12.0 to 30.0 seconds."

Similarity = 0.3044
"... The Hypopnea event is documented from 0.0 to 10.0 seconds. ... The Hypopnea event is
  documented from 10.0 to 30.0 seconds."

================================= Extra Example 1=================================
```

```
Similarity = 0.4007 (original)
"Sleep stage analysis indicates Stable Light Sleep (N2) for this epoch. The Central Apnea event is
   identified from 0.0 to 23.9 seconds. We observe a Oxygen Desaturation event between 29.3 and
   30.0 seconds."

Similarity = 0.2440
"... The Central Apnea event is identified from 0.0 to 6.0 seconds. ..."

Similarity = 0.3053
"... The Central Apnea event is identified from 0.0 to 12.0 seconds. ..."

Similarity = 0.3525
"... The Central Apnea event is identified from 0.0 to 18.0 seconds. ..."

Similarity = 0.3568
"... The Central Apnea event is identified from 0.0 to 24.0 seconds. ..."

Similarity = 0.3326
"... The Central Apnea event is identified from 0.0 to 30.0 seconds. ..."

===================================== Extra Example 2=====================================

Similarity = 0.3294 (original)
"Sleep stage assessment shows Stable Light Sleep (N2) for this epoch. We detect a Arousal event
   from 15.0 to 30.0 seconds."

Similarity = 0.1243
"... We detect a Arousal event from 0.0 to 30.0 seconds."

Similarity = 0.2723
"... We detect a Arousal event from 5.0 to 30.0 seconds."

Similarity = 0.3294
"... We detect a Arousal event from 15.0 to 30.0 seconds."

Similarity = 0.2248
"... We detect a Arousal event from 25.0 to 30.0 seconds."

Similarity = 0.2290
"... We detect a Arousal event from 29.0 to 30.0 seconds."
```

