# OpenReview forum: "SleepLM: Natural-Language Intelligence for Human Sleep"
_ICML.cc/2026/Conference — ICML 2026 spotlight_

### Official Review · Reviewer_Eyyj · 2026-02-23

**Soundness:** 3
**Presentation:** 4
**Significance:** 4
**Originality:** 3
**Overall Recommendation:** 5
**Confidence:** 4

**Summary:**

The paper introduces SleepLM, to the best of my knowledge the first family of sleep-language foundation models that help perceive human sleep through natural language. SleepLM is trained on over 100K hours of PSG data from more than 10,000 individuals, encompassing a large-scale sleep-text dataset. The model jointly learns language and physiological representations using a combination of contrastive alignment, caption generation, and reconstruction losses. A diverse set of experiments on real-world sleep datasets shows that SleepLM achieves strong performance in zero-shot and few-shot learning, cross-modal retrieval, and sleep captioning, beating SOTA baselines. In addition, SleepLM unlocks new capabilities such as generalization to unseen concepts and language-guided event localization.

**Compliance With Llm Reviewing Policy:**

Affirmed.

**Final Justification:**

The paper's rebuttal addressed each of the concerns I had raised previously with detailed explanations and results, which has reinforced my prior assessment and led to an increase in the score from weak accept to accept. Specifically, instead of training for random seeds, the authors justified robustness via bootstrapping, which is valid. Additionally, they provided demographic information, which I requested for, and further broke down results by groups. Overall, the results support the authors' comprehensive experiments and a score raise is well-justified.

**Key Questions For Authors:**

1) Can the authors provide more detailed information about patient cohorts in each dataset, such as age distributions and cohort sizes?
2) The phrase “existing sleep foundation models remain unimodal…” in Section 2 is not entirely true. While the cited papers only use physiological signals, they still comprise multimodal signals like EEG, EOG, etc. Could the authors clarify this phrase better?
3) Are the authors able to provide a percentage breakdown of each task’s classes? It would be helpful to see the distribution of classes for each task to get more context for the F1 and recall scores.
4) For Table 1, “Results are averaged over the SHHS (internal), MrOS (internal), and CFS (external) evaluation sets.” Is there a specific reason for averaging across all datasets? It would be helpful to gauge how the model did separately on internal and external sites than averaging all three.

**Limitations:**

yes.

**Strengths And Weaknesses:**

Strengths:
1) The paper introduces the first sleep-language foundation model and demonstrates that it achieves strong performance across a variety of tasks, while allowing the model to generate new abilities like strong performance on unseen concepts or language-guided localization.
2) The paper presents a detailed comparison of its method against state-of-the-art VLMs and LLMs, which highlights the wide-variety of the experiments conducted by the authors to compare their method against. SleepLM consistently outperforms these existing models and, impressively, proprietary LLMs even when they are given tabular PSG descriptors for sleep stages and event identification.
3) SleepLM demonstrates strong performance on external datasets (CFS and WSC). This demonstrates the strong generalizability of SleepLM to unseen datasets and reduces concerns about the model overfitting solely to its training datasets.
4) The paper conducts in-depth ablations with respect to different model scales (T, S, B), other pretraining variants (Cap, CoCa, CLIP), the effect of removing sleep reconstruction, etc. SleepLM still performs strongly in all these cases.
5) The paper is clearly written, well structured, and easy to follow with informative visuals despite its breadth.


Weaknesses:
1) The results, while impressive, are not fully robust. An average of the scores with standard deviations across multiple seeds would be helpful instead of results from a single run.
2) The paper does not provide detailed information about patient demographics (e.g. patient age distributions) for the datasets. Without this information, it cannot be inferred the exact kind of patient cohort SleepLM generalizes well to (e.g. older patients, pediatrics, etc), which could provide key information on what patient settings in particular does SleepLM perform well.
3) The paper does not provide deployability or efficiency analysis (e.g. latency, inference cost). While the authors acknowledge the paper is a research prototype, understanding computational costs and model deployability analysis can gauge the feasibility of real-time deployment of the model.
4) While the paper explores several ablations on representative tasks, it doesn’t explore comparisons with including reconstruction (e.g., Re + Cap, Re + CLIP) combined with the other methods, which could further isolate the contribution of reconstruction.

---

> ### Author Rebuttal · Authors · 2026-03-31
>
> We sincerely thank the reviewer for the thoughtful feedback and for recognizing our contributions. We hope the detailed responses below address your concerns and further strengthen our submission.
> ***
> > *W1: The results … are not fully robust …*
>
> We appreciate the suggestion. Because pretraining across different seeds is computationally prohibitive, we address your concern via **bootstrapping evaluation** by randomly subsampling 500 samples with replacement for 20 times:
>
> ||Stage|C. Apnea|Hypopnea|Oxy. Desat.|Arousal|Avg Stat|
> |-|-|-|-|-|-|-|
> |Metric|AUC|AUC|AUC|AUC|AUC|sMAPE|
> |SHHS|86.2 ± 1.3|68.6 ± 9.4|76.6 ± 2.0|70.9 ± 2.3|84.4 ± 2.1|2.37 ± 0.25|
> |MROS|82.8 ± 1.4|66.3 ± 8.6|62.9 ± 2.6|66.6 ± 2.0|82.1 ± 2.1|2.76 ± 0.26|
> |CFS|83.3 ± 1.4||77.9 ± 2.3|68.6 ± 2.5|85.6 ± 2.1|3.99 ± 0.36|
>
> As shown, the std across 20 eval sets remain consistently low. This stability confirms that our reported metrics are not artifacts of a specific split.
> ***
> > *W2&Q2: ..does not provide…patient demographics…*
>
> Thank you for raising this. Below, we provide the age distributions for our datasets:
>
> |Train|SHHS (%)|MROS (%)|CCSHS (%)|All (%)|
> |-|-|-|-|-|
> |0-18|0.0|0.0|78.1|3.3|
> |18-35|0.0|0.0|21.9|0.9|
> |35-50|12.9|0.0|0.0|8.4|
> |50+|87.1|100.0|0.0|87.3|
>
> |Val|SHHS (%)|MROS (%)|CFS (%)|All (%)|
> |-|-|-|-|-|
> |0-18|0.0|0.0|20.7|7.3|
> |18-35|0.0|0.0|21.0|7.4|
> |35-50|11.0|0.0|26.5|14.2|
> |50+|89.0|100.0|31.7|71.1|
>
> This demographic skew is expected and aligns with *real world clinical baselines*, as the prevalence of sleep disorders dramatically increases with advanced age [1]. Furthermore, we evaluated the performance stratified by age group on CFS:
>
> ||Stage (AUC)|Hypopnea (IoU)|Oxy. Desat. (IoU)|Arousal (IoU)|
> |-|-|-|-|-|
> |0-18|82.5|12.4|5.3|37.1|
> |18-35|87.0|24.9|20.3|41.7|
> |35-50|82.4|26.5|19.8|36.5|
> |50+|81.1|32.0|17.5|43.7|
>
> SleepLM maintains **robust performance** across age groups. The only minor exceptions are hypopnea and oxy. desat. in the 0-18 cohort. This is again expected, as pediatric respiratory events are the rarest both in our train data and clinical populations [2, 3].
>
> These results confirm that SleepLM generalizes well and does not overfit to older patient profiles. We will include these analyses in the revision.
> ***
> > *W3: The paper does not … efficiency analysis*
>
> Thank you for the feedback, we have profiled SleepLM during inference. Please see **Table 9** for a detailed breakdown of the model parameters. SleepLM maintains competitive inference overhead. Zero-shot inference requires **268 GFLOPs**. For fewshot, the sleep encoder (118 GFLOPs) is close to SSL baselines (99 GFLOPs).
>
> These results suggest that SleepLM is capable of real-time deployment on standard consumer hardware. We will include a detailed discussion of these considerations.
> ***
> > *W4: …isolate the contribution of reconstruction*
>
> We agree additional ablations better isolate reconstruction's contribution. Since we propose reconstruction to resolve the information density mismatch between signals and text during contrastive alignment, we performed new ablations on CLIP for sleep staging:
>
> |CLIP|SHHS|MROS|CFS|
> |-|-|-|-|
> |w/o Recon.|85.6|83.5|83.0|
> |Recon.|85.9|84.2|83.0|
>
> As shown, reconstruction improves performance on two of the three datasets.
> ***
>  > *Q2: ..unimodal…clarify this phrase better?*
>
> Many thanks for the catch. By 'unimodal,' we meant limited to the physiological domain. To avoid confusion, we will update Section 2 to clarify this clearly.
> ***
> > *Q3: …percentage breakdown of each task’s classes? …*
>
> Thank you for the insightful question. Due to the character limit, we respectfully direct the reviewer to our comprehensive response to [Reviewer PKR9( Sound2&Q2)](https://openreview.net/forum?id=9wpwfSJCp9&noteId=NRwAvlZiDC) for the detailed breakdown.
>
> As noted, our distribution reflects natural physiological prevalence. Our evaluation accounts for this imbalance, proving SleepLM is robust across all classes
> ***
>  > *Q4: …how the model did separately on internal and external sites … *
>
> We agree this breakdown is vital. Please refer to **Appendix D.5** for our unaggregated results per dataset. As shown, SleepLM maintains strong performance across both internal and external sites, demonstrating robust generalization to unseen clinical settings.
> ***
> Reference:
> 1. Increased prevalence of sleep-disordered breathing in adults. 2013.
> 2. Pathophysiology of upper airway obstruction... 2004.
> 3. Obstructive sleep apnea syndrome in children. 2011.
> 4. Meta-analysis of quantitative sleep parameters... 2004.
> 5. Treatment of central sleep apnea in adults... 2025.
> ***
> We sincerely thank the reviewer once again for their time and constructive feedback. To support the community and enable future research, we remain committed to fully open-sourcing our research. We hope our detailed responses and the additional experiments have adequately addressed your concerns, and we would be very grateful if you might consider raising your score in light of these updates.

---

> > ### Author Rebuttal · Reviewer_Eyyj · 2026-04-04
> >
> > Thanks, my concerns have been fully resolved and I have changed my scores accordingly.

---

> > > ### Author Response · Authors · 2026-04-07
> > >
> > > We are grateful for your support and positive remark! Your recommendations to include detailed patient demographics and exact class distributions were constructive points that have helped strengthen the paper, and these updates will be outlined in the revised manuscript. We are happy to answer any remaining questions you might have.

---

### Official Review · Reviewer_EcGQ · 2026-03-11

**Soundness:** 4
**Presentation:** 3
**Significance:** 3
**Originality:** 3
**Overall Recommendation:** 5
**Confidence:** 4

**Summary:**

Analyzing sleep has been a difficult task due to its predefined stages and events which limit generalization to sleep phenomena. The authors propose a unity between natural language and multimodal polysomnographical models, allowing robust representations of sleep physiology. The authors also curate the first large sleep-scale dataset for this model with unified generation pipeline and pretraining objectives. The authors report better than current SASO zero- and few-shot learning models in sleep tasks.

**Compliance With Llm Reviewing Policy:**

Affirmed.

**Final Justification:**

With the inclusion of direct comparisons to previous sleep models such as SleepFM, as well as further expansion upon the utilities and valid input data, I am confident in my belief that this paper should be accepted. There is very clear novelty, building upon an established field of sleep foundation models, which show great improvements compared to the previous iteration of sleep models. The presentation and clarity of the paper is very high, and all questioned parts of the experimental setup and data preparation are highly detailed and justified.

**Key Questions For Authors:**

1. Sleep foundation models without language have been explored by Thapa et al., 2024 with the presentation of SleepFM, but I did not see a direct comparison in performance between SleepLM and SleepFM. If you do have this comparison, would you be able to inform me of SleepFM's performance on the tasks in this manuscript, or point me to where this comparison was made? If you do not, could you explain why this was not done?

2. Is SleepLM compatible with datasets not including sleep-text? I.e. in Thapa et al., 2024 where they evaluate SleepFM against four separate sleep datasets, would SleepLM also be able to take these datasets and perform the same tasks as evaluated in that publication?

3. Are there other means to verify the sleep-text corpus for validity? It is difficult for me to properly judge the results of the models on this dataset without additional means of data integrity or verification.

**Limitations:**

yes

**Strengths And Weaknesses:**

1. Novel framework. This is the first case of a language-integrated sleep foundation, which opens the door to multiple new frames of analyses, especially with existing datasets.
2. Performance compared to baseline. SleepLM outperforms all SASO VLM and LLMs for eight out of nine performance metrics, four out of five sleep tasks (Sleep Stage, Event, HR, and Channel Stats,) with SpO2 having significantly higher recall with 15% worsened MAE. Though MAE vs MSE vs corr or other metrics is called into question, the degree to which all other metrics increase relative to how MAE worsens, this is not as crucial.
3. Thorough analyses and theoretical basis. Each step and segment of SleepLM is documented and supported either with previous publications or mathematical groundings, along with clear explanations of how improvements should present themselves during post-training scoring, which is supported in figures (like figure 4.)

Weaknesses:
1. No comparison to previous sleep foundation models. Unless I have misunderstood the tables, SleepLM is only compared to other SASO LLMs and VLMs instead of a proposed foundation model without language (such as SleepFM.) It would be greatly beneficial to the impact of the paper if SleepLM was directly compared against SleepFM to see how much language actually helps - this could also be done in an ablation study removing the language component.
2. Limited datasets. While the authors present the new curated dataset for usage in this field, it remains to be seen how SleepLM can perform in tasks meant for precursor sleep foundation models after being trained to take in language. This would also help to see to what degree language (or sleep data) is impacting the model's learning, or if the sum of the parts would be greater than the whole, in this case.

---

> ### Author Rebuttal · Authors · 2026-03-31
>
> We sincerely thank the reviewer for their constructive feedback and positive evaluation of our work. We hope the detailed responses below address your concerns and further strengthen our submission.
> ***
> > *W1&Q1: No comparison to previous sleep foundation models…*
>
> We appreciate this suggestion. Our original evaluation primarily focused on SleepLM’s novel *zero-shot capabilities* (retrieval, generation). Conversely, SleepFM and other existing domain specific methods are *fully/self supervised* and need  finetuning task-specific heads for downstream tasks [1, 2, 3, 6], restricting them to fewshot rather than true zeroshot settings.
>
> But we agree a direct comparison is crucial. So we conduct fewshot linear probing experiments against 3 types of domain specific models: **(1)** *Supervised sleep staging models* (RobustSleepNet[1]), **(2)** *Sleep FMs* (SleepFM[2]), **(3)** *General time series FMs* (Chronos2[3]) and, when necessary, retraining following our exact settings in **Appendix A. B.** for direct comparison. The K-shot performance is compared on the external CFS dataset:
>
> ||k|Stage AUC|Arousal AUC|Hypopnea AUC|Oxy. Desat. AUC|
> |-|-|-|-|-|-|
> |SleepLM|10|**81.6**|**84.8**|**64.9**|61.5|
> ||50|**90.8**|**90.7**|**85.5**|80.3|
> |SleepFM|10|74.4|84.3|62.6|**65.2**|
> ||50|85.5|87.9|79.9|**83.7**|
> |Chronos-2|10|64.4|75.3|52.6|64.8|
> ||50|64.9|78.3|63.6|68.1|
> |RobustSleepNet|10|81.4|-|-|-|
> ||50|88.7|-|-|-|
>
> As shown in the table, under this fewshot linear probing setting, SleepLM outperforms/matches baseline models across nearly all tasks. This result suggests that we learn strong sleep encoders natively.
>
> We attribute this to: (1) **multilevel language supervision (Table 10)**, which we have shown to improve global semantics, and (2) **ReCoCa architectures (Table 4)**, where extensive ablations show that it is more effective than other common model variants.
>
> We will include these new analysis in our revised manuscript.
> ***
> > *W2&Q2: Is SleepLM compatible with datasets not including sleep-text …*
>
> Thank you for raising this. Yes, SleepLM is fully compatible with datasets that do not contain text. Our signal encoder operates entirely independently of the language module during downstream adaptation. As our linear probing results demonstrate, SleepLM matches SleepFM on standard supervised tasks, retaining core feature extraction capabilities while unlocking new language-driven paradigms*:
>
> |Task|Sleep Foundation Models|SleepLM|
> |-|-|-|
> |Supervised |Yes|Yes|
> |Zero-Shot|No|Yes|
> |Retrieval|No|Yes|
> |Concept Generalization|No|Yes|
>
> By mapping signals into a continuous semantic space with language supervision, SleepLM generalizes to semantically similar but unseen concepts (**Fig. 4**) and enables nuanced retrieval along semantic gradients rather than exact categorical matches (**Fig. 8**).
>
> Regarding the datasets in Thapa et al., while the brief rebuttal window prevent processing entirely new data, we would like to note that MrOS is already a core evaluation dataset (**Table 1, 12**). Furthermore, our new linear probing experiments demonstrate highly competitive performance when compared against SleepFM and other domain specific baselines.
>
> We will explicitly highlight this capability comparison in our revision.
> ***
> > *Q3: Other means…verify the sleep-text corpus for validity?*
>
> Thank you for asking this. We want to clarify that our captions are constructed through a highly controlled, deterministic pipeline grounded in established clinical truth:
>
> **-Expert Annotations**: High level semantic labels are drawn from the human expert.
>
> **-Calibrated Algorithms**: Low level stats details are extracted using well-established algorithms.
>
> **-Diverse Paraphrasing**: We render these facts into diverse templates from LLMs, which introduces lexical variety while preventing hallucinations.
>
> For structural verification, we provide the pipeline visualization in **Fig. 2** and random caption samples in **Appendices E.1, E.4**.
>
> In addition, we employ strict **subject-level splitting** across **gold standard cohorts** in the field [1, 4, 5]. To guarantee zero data leakage and valid test generalization, we completely held out external datasets (CFS, WSC) during pretraining. SleepLM maintains strong performance with minimal degradation on these entirely unseen cohorts.
> ***
> Reference
> 1. RobustSleepNet: Transfer learning... 2021.
> 2. SleepFM: multi-modal representation learning... 2024.
> 3. Chronos-2: From univariate... 2025.
> 4. Scaling up scientific discovery in sleep medicine... 2016.
> 5. U-Sleep: resilient high-frequency sleep staging. 2021.
> ***
> We sincerely thank the reviewer once again for their time and constructive feedback. To support the community and enable future research, we remain committed to fully open-sourcing our research. We hope our detailed responses and the additional experiments have adequately addressed your concerns, and we would be very grateful if you might consider raising your score in light of these updates.

---

> > ### Author Rebuttal · Reviewer_EcGQ · 2026-04-03
> >
> > Thank you for this well-detailed response. You have address every one of my concerns with great detail and thoroughness. My scores will be updated accordingly.

---

> > > ### Author Response · Authors · 2026-04-07
> > >
> > > Thank you for your encouraging feedback! We are glad to learn that our response has addressed your concerns, and we sincerely appreciate your willingness to update the score. Once again, thanks for the time and insightful comments that have helped improve our manuscript.

---

### Official Review · Reviewer_EdLw · 2026-03-13

**Soundness:** 2
**Presentation:** 3
**Significance:** 3
**Originality:** 3
**Overall Recommendation:** 4
**Confidence:** 4

**Summary:**

This paper proposes SleepLM, a sleep language foundation model that aligns polysomnography (PSG) signals with natural language. The authors introduce a multi-level caption generation pipeline to produce textual descriptions and design a multimodal training architecture (ReCoCa) that integrates contrastive learning, text generation, and signal reconstruction. Experiments on multiple sleep-related tasks show that the proposed method performs well in zero-shot recognition, cross-modal retrieval, and few-shot learning tasks.

**Compliance With Llm Reviewing Policy:**

Affirmed.

**Final Justification:**

Most of my doubts have been resolved, but overall, I will maintain my score.

**Key Questions For Authors:**

1.For non-text tasks such as classification, why not use more specialized baselines, such as time-series foundation models or dedicated sleep analysis models? Please add relevant experiments to better demonstrate the capability of SleepLM.

2.Although the authors state in the paper that overly long epochs make the use of proprietary LLMs extremely expensive in terms of both computation and financial cost, why not compare SleepLM against itself under different input lengths? For example, please add experiments evaluating SleepLM with epoch sizes of 2,000, 20,000, and 200,000.

3.Since Figure 1 mentions the METEOR score, why are the corresponding experimental results not reported in the paper? Please include the relevant experiments.

4.To what extent do these textual descriptions introduce information that cannot be directly inferred from the signals themselves?

**Limitations:**

See above

**Strengths And Weaknesses:**

Strengths

1.This paper explores a direction with potentially significant impact: aligning physiological time-series signals with natural language.

2.The experimental design is relatively comprehensive, covering tasks such as zero-shot recognition, cross-modal retrieval, unseen concept generalization, and few-shot transfer.

3.The proposed model integrates multiple training objectives into a unified framework, and ablation studies demonstrate its performance advantages.

4.The authors introduce a multi-level description generation pipeline tailored for sleep data and construct a large-scale Sleep-Text dataset based on multiple PSG datasets.

Weaknesses

1.The comparison with general-purpose large language models may be unfair

The paper compares SleepLM with general-purpose large language models such as Gemini and DeepSeek. However, general-purpose LLMs are not designed to process raw multivariate time-series signals, whereas SleepLM uses an encoder specifically designed for physiological signals. Therefore, this comparison may not be entirely fair, the baseline models are placed in an inherently disadvantaged setting.

2.The experimental details are insufficient

The selection of baselines appears to have been carefully curated by the authors. There are already many general time-series foundation models and sleep foundation models in the literature, but the paper does not compare against them.

The test set is too small. Each epoch in the test set is 30 seconds, and there are only 2,000 epochs in total, which is equivalent to merely two nights of data. This is not convincing for evaluating generalization.

The proposed model, SleepLM, is capable of performing many tasks beyond text generation, but the authors do not compare it with corresponding state-of-the-art domain-specific models on those tasks.

Table 2 is not meaningful. Due to context length limitations, neither DeepSeek-R1 nor Gemini 2.5 Pro can complete the corresponding experiments. Therefore, I do not think Table 2 provides meaningful evidence.

Figure 1 includes BERTScore and METEOR scores, but the authors do not report or explain their meaning or role in the main text or appendix. In fact, METEOR is not mentioned anywhere else in the paper.

3.The methodological novelty is relatively limited
The proposed ReCoCa framework combines several well-known training objectives:contrastive alignment, caption generation, signal reconstruction, Although this combination is reasonable, each component appears to be derived from existing multimodal training paradigms (e.g., CLIP, CoCa, and masked reconstruction). Therefore, from a methodological perspective, the overall novelty appears relatively limited.

4.The language supervision is automatically generated and may not introduce additional semantic information

A major concern is that the text supervision used in this paper is automatically generated from signal statistics via templates, rather than written by humans as natural language descriptions. Based on the paper’s description of the caption generation pipeline, many of the texts are generated from deterministic signal statistics (e.g., EEG band power, heart rate statistics, respiratory features, etc.). Therefore, these textual supervisions may simply restate information already contained in the signals in linguistic form. This raises the question of whether the model is truly learning semantic alignment between language and physiological signals, or whether it is effectively equivalent to a form of self-supervised learning that converts signal features into text.

5.Formatting and wording issues

There are problems with the numbering of figure and table references. Please check and correct them carefully.
The paper uses LLM to refer to DeepSeek and Gemini, and VLM to refer to Qwen3-VL and LLaVA-NeXT. However, there are places where LLM and VLM are incorrectly used. For example, in Appendix A:
 “Baseline Finetuning Strategy:vFor the multimodal LLM baselines (Qwen3-VL and LLaVA-NeXT)...”
 Here, Qwen3-VL and LLaVA-NeXT should be referred to as multimodal VLM baselines, not LLM baselines, and there is also an extra letter “v” in “vFor.” Please carefully proofread and revise the entire manuscript.

---

> ### Author Rebuttal · Authors · 2026-03-31
>
> We sincerely thank the reviewer for their careful review and constructive feedback. We hope the detailed responses below address your concerns and further strengthen our submission.
> ***
> > *W1: …comparison with…language models…unfair*
>
> Thank you for this observation. We would like to clarify our rationale for selecting general purpose LLMs/VLMs:
>
> A core contribution of SleepLM is performing downstream tasks in a zero shot manner via **direct text generation** or **cross modal retrieval**. As existing domain specific models require supervised heads on task specific labels [1, 2, 3] multimodal models remain the **only** baselines capable of this exact zero shot setting.
>
> To ensure fair comparison we finetuned LLaVA NeXT and Qwen3 VL on our **exact pretraining dataset and settings**. Despite these baselines having orders of magnitude more parameters and a **distinct capacity advantage**, SleepLM still demonstrates superior performance.
>
> We welcome suggestions for additional baselines that might fit this zero-shot setting. To further address your underlying concern, we provide additional evaluation results in our next response.
> ***
> > *W2a,W2c&Q1:…Please add relevant experiments to better demonstrate the capability of SleepLM.*
>
> Due to space limits, we respectfully point the reviewer to **Reviewer PKR9( Sound3&Q1)** for our comparisons against domain-specific models. Briefly, SleepLM matches or outperforms these baselines, confirming a strong sleep embedding.
> ***
> > *W2d: Table 2 is not meaningful…*
>
> We respectfully suggest that the **inability of LLMs to complete this experiment** is what makes the comparison meaningful. A core premise of our work is that standard LLMs cannot handle dense PSG signals due to **context constraints**, a bottleneck empirically demonstrated in Table 2. Furthermore, even on a feasible subset, LLM performance remained near random, proving that efficient retrieval is a **novel capability** uniquely unlocked by SleepLM.
> ***
> > *W4&Q4: …automatically generated…descriptions…information that cannot… inferred from the signals…*
>
> We agree training with free text is ideal. Due to character limits, please see our detailed response to **Reviewer PKR9(Sig2)** regarding our language supervision. Briefly, by scaling diverse, LLM-rephrased templates with expert annotations, we inject **continuous semantic knowledge**. This uniquely unlocks capabilities unachievable via signal-based SSL, such as zero-shot generalization and flexible retrieval.
>
> ***
> > *W2e&Q3: …the METEOR score…Please include the relevant experiments*
>
> Thank you for the catch. We provide the results below and will include it in our revision:
>
> ||BERTPrecision|BERTRecall|BERTF1|METEOR|
> |-|-|-|-|-|
> |DeepSeekR1|89.8|89.7|89.7|32.5|
> |Gemini2.5pro|87.8|88.9|88.3|33.0|
> |LlaVA-Next|91.2|90.5|90.8|42.3|
> |SleepLM|**91.8**|**92.0**|**91.8**|**44.2**|
>
> As shown, SleepLM excels across all metrics by generating accurate physiological text.
> ***
> > *W2b&Q2: The test set is too small … not convincing for evaluating generalization.*
>
> We agree that demonstrating robust generalization at scale is critical. To address this, we expanded our evaluation:
>
> **Scaled Evaluation**: As shown below, SleepLM's performance remains highly stable even when scaled to 20,000 and 50,000 epochs on the CFS dataset.
>
> ||Stage|Hypopnea|Arousal|Oxy. Desat.|Avg Stat|
> |-|-|-|-|-|-|
> ||AUC|BAcc|BAcc|BAcc|sMAPE|
> |2k|83.7|78.1|86.9|69.1|3.99|
> |20k|84.1|78.8|85.2|67.6|4.24|
> |50k|84.5|78.6|85.5|67.9|4.22|
>
> **Test Diversity**: Though computational constraints limit our validation set to 2,000 epochs, these are not continuous blocks from just two individuals. To ensure high representativeness, we strictly stratified this sample across **930 distinct patients (SHHS: 409, MrOS: 193, CFS: 328)**, capturing a truly diverse population.
>
> **Bootstrapping Analysis**: Due to space limits, please see **Reviewer Eyyj(W1)** for our full bootstrapping results. In short, it exhibits low variance across all datasets, confirming our reported performance is not an artifact of random split.
>
> Together, these results prove the stability of our evaluation.
> ***
> > *W5: Formatting and wording issues*
>
> Thank you for the meticulous reading. We will definitely correct the typos and references to incorporate your detailed suggestions.
> ***
> Reference:
> 1. SleepFM: multi-modal representation... 2024.
> 2. RobustSleepNet: Transfer learning... 2021.
> 3. Chronos-2: From univariate... 2025.
> ***
> We sincerely thank the reviewer once again for their time and constructive feedback. To support the community and enable future research, we remain committed to fully open-sourcing our code, models, checkpoints, and datasets. We hope our detailed responses and the additional experiments have adequately addressed your concerns, and we would be very grateful if you might consider raising your score in light of these updates.

---

> > ### Author Rebuttal · Reviewer_EdLw · 2026-04-04
> >
> > The authors’ response does not address my core concern and instead avoids the key issue. What I need is a comparison between SleepLM and specialized small models on downstream tasks, rather than further emphasis on SleepLM’s few-shot capabilities. Even if SleepLM underperforms compared to task-specific models, that would still be acceptable—as long as the authors can demonstrate that SleepLM achieves reasonably competitive performance on specific tasks. Therefore, I will maintain my current score.

---

> > > ### Author Response · Authors · 2026-04-07
> > >
> > > Thank you for your feedback. We apologize if our previous response caused any confusion. We would like to clarify this point more directly.
> > >
> > > A primary contribution of SleepLM is its zero-shot inference capability, requiring *no downstream labels or parameter updates*. In contrast, existing supervised/self-supervised domain-specific models are not designed for this setting. To perform downstream tasks, they fundamentally require either (1) supervised training/finetuning *with downstream labels*, or (2) linear-probing with a task-specific head, again *on downstream labels*.  As a result, directly comparing SleepLM’s zero-shot results against finetuned task-specific models would not be an apples-to-apples comparison. This is why, in our earlier response, we included few-shot linear-probing as a more comparable and computationally efficient evaluation.
> > >
> > > To fully address the reviewer’s concern, we conducted additional experiments to cover all plausible settings.
> > >
> > > As before, we aim to cover a diverse set of baselines to ensure a unified comparison. For supervised baselines, we selected RobustSleepNet and U-Sleep because they are established and widely used [1]. For event (e.g. apnea) detection, strong task-specific baselines are scarce: most estimate overall severity (e.g., AHI) rather than dense localization [2, 3], while others use limited inputs and lack reproducible code [4]. Thus, among public and reproducible models, we were only able to identify versatile sleep staging baselines suitable for a valid comparison.
> > >
> > > **Setting #1: Full finetuning**
> > >
> > > To compare against domain-specific models on their native tasks, while these specialized models are designed for a *'pretrain-then-finetune'* paradigm, we evaluated SleepLM under the **exact conditions** for the fairest comparison.
> > >
> > > Specifically, we pretrained these models using the same splits as SleepLM. We then isolated SleepLM’s sleep encoder and **fully finetuned** all models on the training split of the unseen CFS dataset. Final AUCs are reported on the full test split. For Chronos-2, we finetuned its public pretrained checkpoint to appropriately assess it as a general time-series FM.
> > >
> > > ||Stage|Arousal|Hypopnea|Oxy. Desat.|
> > > |-|-|-|-|-|
> > > |SleepLM|87.2|**89.2**|83.0|**84.1**|
> > > |Chronos-2|**87.6**|87.9|**85.0**|83.7|
> > > |SleepFM|85.4|86.7|81.7|80.3|
> > > |RobustSleepNet|85.8|-|-|-|
> > > |U-Sleep|83.8|-|-|-|
> > >
> > > &nbsp;
> > >
> > > **Setting #2: Zero-shot inference**
> > >
> > > Because domain-specific models require a task-specific head, true zero-shot evaluation is impossible. As a fair proxy, we trained supervised domain-specific baselines on the same splits for SleepLM pretraining, using the corresponding ground-truth labels, and then evaluated them directly on CFS without finetuning.
> > >
> > > (Note: fully training larger baselines exceeds the computational limits of the rebuttal period. SleepFM and Chronos-2 results for this setting will be included in the revision)
> > >
> > > ||BAcc|F1|AUC|
> > > |-|-|-|-|
> > > |SleepLM (zero-shot)|74.2|69.5|83.7|
> > > |RobustSleepNet|60.1|57.0|75.3|
> > > |U-Sleep|74.0|69.1|82.9|
> > >
> > > &nbsp;
> > >
> > > **Setting #3: Few-shot**
> > >
> > > As presented previously, we conducted few-shot linear probing by freezing pretrained encoders and training only a linear head on limited downstream data. We updated our earlier results to report binarized AUC instead of probability-based AUC for consistency and easier comparisons:
> > >
> > > ||k|Stage|Arousal|Hypopnea|Oxy. Desat.|
> > > |-|-|-|-|-|-|
> > > |SleepLM|10|**71.1**|66.5|59.3|**62.3**|
> > > ||50|**79.9**|**79.5**|**75.1**|**76.8**|
> > > |Chronos-2|10|59.6|64.6|50.1|60.0|
> > > ||50|60.4|68.3|65.1|66.8|
> > > |SleepFM|10|64.9|**70.2**|**60.4**|61.8|
> > > ||50|74.7|76.8|74.0|72.8|
> > > |RobustSleepNet|10|69.1|-|-|-|
> > > ||50|76.4|-|-|-|
> > > |U-Sleep|10|61.8|-|-|-|
> > > ||50|68.9|-|-|-|
> > >
> > > Overall, SleepLM remains competitive across tasks and settings. Also, generally FMs perform better than smaller specialized models. We attribute this in part to the substantially **larger parameter scale** of FMs relative to these specialized baselines. This likely stems from two factors: (1) the established scaling behavior with model size that we also observe for SleepLM (**Table 8, Appendix D.2**), and (2) the ability of larger models to learn representations that transfer better across tasks and datasets. This is further supported by the strong performance of Chronos-2 in our full fine-tuning experiments. We provide their model sizes below:
> > >
> > > |||
> > > |-|-|
> > > |SleepLM|50M|
> > > |Chronos-2|120M|
> > > |SleepFM|10M|
> > > |RobustSleepNet|100K|
> > > |U-Sleep|500K|
> > > ---
> > > Reference:
> > > 1. U-Sleep: resilient high-frequency sleep staging, 2021
> > > 2. Deep learning-based event counting for apnea-hypopnea…, 2024
> > > 3. A Comprehensive study on a deep-learning-based…, 2024
> > > 4. Deep learning of sleep apnea-hypopnea events…, 2024
> > >
> > > &nbsp;
> > >
> > > ---
> > > We hope these results and clarifications have fully addressed your concerns. Thank you again for your time and constructive feedback. If the reviewer feels the concern has now been adequately addressed, we would be grateful if they would consider updating their score accordingly.

---

### Official Review · Reviewer_PKR9 · 2026-03-13

**Soundness:** 3
**Presentation:** 3
**Significance:** 3
**Originality:** 3
**Overall Recommendation:** 5
**Confidence:** 3

**Summary:**

The authors propose integrating sleep signals with natural language by generating text captions from annotations (sleep stages, apnea) and automated feature extraction (HR, HRV, power bands, respiration rate) at three levels: per-channel, temporally local, and global window. They introduce ReCoCa, which contrastively aligns signal and caption embeddings while jointly performing signal and caption reconstruction. The model is shown to outperform existing LLMs and fine-tuned VLMs at crossing the signal-text barrier, and can zero-shot identify correlated unseen concepts like mixed and obstructive apnea. No comparison to sleep-specific models is made.

**Compliance With Llm Reviewing Policy:**

Affirmed.

**Final Justification:**

The paper presents the first demonstration of sleep signal–language integration at both segment and full-recording level, which is a genuine novel contribution. The natural language formulation enabling zero-shot prediction of unseen concepts is the key strength. My main concerns were: (1) the absence of comparisons to sleep-specific models for staging and apnea, (2) unreported class distributions making results hard to interpret, and (3) the limitation of relying entirely on auto-generated captions. The rebuttal addressed all: few-shot linear probing against RobustSleepNet, SleepFM, and Chronos-2 shows SleepLM matching or outperforming domain-specific baselines, class distributions were provided and confirm that imbalanced classes reflect natural prevalence rather than a design flaw, and the authors acknowledged the caption limitation with a commitment to discuss it explicitly. I maintain my recommendation of Accept, with the expectation that the revised manuscript includes the new baselines, class distributions, and a discussion of the auto-generated caption limitation.

**Key Questions For Authors:**

- Why is there no comparison to sleep-specific models for staging and apnea? This is needed to understand where ReCoCa actually stands.
- What is the class distribution for sleep stages and apnea events in the dataset? Without this, reported performance is hard to interpret.

**Limitations:**

Yes, but please add note on auto-generated caption dataset.

**Strengths And Weaknesses:**

Soundness:

- Clear improvement over existing LLMs and fine-tuned VLMs with ablations across multiple dimensions
- On the sampled validation set class distribution for  sleep stages and apnea events is not reported, making it hard to interpret the reported performance
- No comparison to high-performance sleep-specific models for staging or apnea

Presentation:

- Well written and easy to follow

Significance:

- Natural language formulation enables zero-shot prediction of unseen concepts, opening up many future analysis possibilities for sleep data
- The dataset is based entirely on automated annotations and feature extraction, with no clinician notes, which limits practical clinical utility

Originality:

- First demonstration of sleep signal-language integration at both segment and full-recording level in sleep, which is a genuine novel contribution

---

> ### Author Rebuttal · Authors · 2026-03-31
>
> We thank the reviewer for the thoughtful feedback and for recognizing the originality of our contribution. We hope the detailed responses below address your concerns and strengthen our submission.
> ***
> > *Sound2&Q2: …What is the class distribution…*
>
> Thank you for the suggestion. We agree that the class distribution is critical for interpreting the results. We provide it across all validation datasets:
>
> ||SHHS (%)|MROS (%)|CFS (%)|All (%)|
> |-|-|-|-|-|
> |Wake|18.4|24.6|21.6|21.5|
> |N1|6.6|7.8|5.9|6.7|
> |N2|42.1|43.0|40.7|42.0|
> |N3|14.3|8.6|15.1|12.7|
> |REM|18.6|15.9|16.8|17.1|
> |C. Apnea|2.4|1.8|0.0|1.4|
> |Hypopnea|34.8|17.1|21.6|24.5|
> |Oxy. Desat.|29.0|54.0|22.9|35.3|
> |Arousal|22.9|24.1|21.3|22.8|
>
> We note that N1 and Central Apnea are rare. This accurately reflects their **natural physiological prevalence**. For instance, N1 typically accounts for ~5% of total sleep time in healthy individuals [1], and C. Apnea prevalence is also much lower compared to other Resp. events [2].
>
> Because we anticipate this imbalance, all evaluations are performed with **multiple metrics** that are designed to handle data imbalance (ex., BAcc., AUC) to make sure our reported performance is robust across all classes (**Table 1, 3, 12**).
>
> We will add this information in our revision.
> ***
> > *Sound3&Q1: Why is there no comparison to sleep-specific models…*
>
> Thank you for raising this. We agree that including domain-specific models is crucial. We conduct additional experiments below to support our claims.
>
> First, we note that existing domain specific models are either *fully supervised* or *self-supervised* models that require training an additional task-specific head with downstream labels [3, 4, 5]. This means they are **only** able of *few-shot* predictions (rather than zero-shot recognition / retrieval / generation as SleepLM).
>
> Therefore, to address your concern, we conduct few-shot linear probing runs against diverse types of domain specific models: **(1)** *Supervised sleep stage models* (RobustSleepNet [3]), **(2)** *Sleep FMs* (SleepFM [4]), and **(3)** *General time series FMs* (Chronos2 [5]). When necessary,  models are retrained using the exact same settings in **Appendix A. B.** for direct comparisons on the external CFS dataset:
>
> ||k|Stage AUC|Arousal AUC|Hypopnea AUC|Oxy. Desat. AUC|
> |-|-|-|-|-|-|
> |SleepLM|10|**81.6**|**84.8**|**64.9**|61.5|
> ||50|**90.8**|**90.7**|**85.5**|80.3|
> |SleepFM|10|74.4|84.3|62.6|**65.2**|
> ||50|85.5|87.9|79.9|**83.7**|
> |Chronos-2|10|64.4|75.3|52.6|64.8|
> ||50|64.9|78.3|63.6|68.1|
> |RobustSleepNet|10|81.4|-|-|-|
> ||50|88.7|-|-|-|
>
> As shown in the table, under this few-shot linear probing setting, SleepLM outperforms/matches baseline models across nearly all tasks. This result suggests that we learn strong sleep encoders out of the box.
>
> We attribute this to: (1) **multilevel language supervision (Table 10)**, which we have shown to improve global semantics, and (2) **ReCoCa architectures (Table 4)**, where extensive ablations show that it is more effective than other common model variants.
>
> We will include these new baselines and the accompanying analysis in our revised manuscript.
> ***
> > *Sig2: …automated annotations…limits practical clinical utility*
>
> Thank you for the feedback, and we agree authentic clinician notes would be ideal. But such info are *rare*, *unstructured*, *labor intensive* to acquire at scale,  and currently unavailable in any existing open source, large sleep datasets.
>
> Despite this, our language supervision surpasses discrete labels and enables unique utilities: (1) **Unseen Generalization**: Mapping physiological signals into a continuous semantic space enables generalization to novel concepts zeroshot (**Fig. 4**). (2) **Semantics Retrieval**: The continuous physiological states allow explorations based on semantic gradients than exact categorical matches, opening new avenues for *educational training and research discovery* (**Fig. 8**). (3) **Precise Event Segmentation**: Temporally grounded representations enable precise, language-guided localization (**Fig. 7**).
>
> We view our work as an initial effort to connect natural language with sleep. We will explicitly discuss the integration of authentic clinician notes as a critical future direction in our revised manuscript.
> ***
> Reference
> 1. Meta-analysis of quantitative sleep parameters from childhood to old age… 2004.
> 2. Treatment of central sleep apnea in adults... 2025.
> 3. RobustSleepNet: Transfer learning... 2021.
> 4. SleepFM: multi-modal representation learning... 2024.
> 5. Chronos-2: From univariate to... 2025.
> ***
> We sincerely thank the reviewer once again for their time and constructive feedback. To support the community and enable future research, we remain committed to fully open-sourcing our research. We hope our detailed responses and the additional experiments have adequately addressed your concerns, and we would be very grateful if you might consider raising your score in light of these updates.

---

> > ### Author Rebuttal · Reviewer_PKR9 · 2026-04-02
> >
> > Good rebuttal - I will keep my score.

---

> > > ### Author Response · Authors · 2026-04-07
> > >
> > > Thank you for your encouraging feedback and confirmation that our rebuttal has fully addressed your concerns! Your constructive comments, especially regarding sleep specific baselines and our text dataset limitations, was highly valuable and will be incorporated into the revised manuscript. We remain available should any further questions arise.

---

### Decision · Program_Chairs · 2026-04-30

**Decision:**

Accept (spotlight)

**Comment:**

This paper introduces SleepLM, the first foundation model to connect complex sleep signals with natural language. By building a large-scale sleep-text dataset and a specialized architecture (ReCoCa), the authors have unlocked a brand new capabilities for the field. While reviewers initially raised concerns regarding the lack of domain-specific baselines, the authors' rebuttal successfully addressed these points. Reviewers significantly raised their scores following these clarifications, highlighting the technical soundness and high impact of the first large-scale sleep-language foundation model. AC finds the paper technically solid and a valuable contribution to the ICML 2026 community.